RESEARCH ARTICLE  

# Spike frequency adaptation supports network computations on temporally dispersed information

**Darjan Salaj[1†], Anand Subramoney[1†], Ceca Kraisnikovic[1†], Guillaume Bellec[1,2], Robert Legenstein[1], Wolfgang Maass[1]\***

[1]Institute of Theoretical Computer Science, Graz University of Technology, Graz, Austria; [2]Laboratory of Computational Neuroscience, Ecole Polytechnique Fédérale de Lausanne (EPFL), Lausanne, Switzerland

**Abstract** For solving tasks such as recognizing a song, answering a question, or inverting a sequence of symbols, cortical microcircuits need to integrate and manipulate information that was dispersed over time during the preceding seconds. Creating biologically realistic models for the underlying computations, especially with spiking neurons and for behaviorally relevant integration time spans, is notoriously difficult. We examine the role of spike frequency adaptation in such computations and find that it has a surprisingly large impact. The inclusion of this well-known property of a substantial fraction of neurons in the neocortex – especially in higher areas of the human neocortex – moves the performance of spiking neural network models for computations on network inputs that are temporally dispersed from a fairly low level up to the performance level of the human brain.

**\*For correspondence:**
maass@igi.tugraz.at

[†]These authors contributed equally to this work

**Competing interests:** The authors declare that no competing interests exist.

## Introduction

Since brains have to operate in dynamic environments and during ego-motion, neural networks of the brain need to be able to solve 'temporal computing tasks', that is, tasks that require integration and manipulation of temporally dispersed information from continuous input streams on the behavioral time scale of seconds. Models for neural networks of the brain have inherent difficulties in carrying out such temporal computations on the time scale of seconds since spikes and postsynaptic potentials take place on the much shorter time scales of milliseconds and tens of milliseconds. It is well known that biological neurons and synapses are also subject to a host of slower dynamic processes, but it has remained unclear whether any of these can be recruited for robust temporal computation on the time scale of seconds. We focus here on a particularly prominent one of these slower processes: spike frequency adaptation (SFA) of neurons. SFA denotes the effect that preceding firing activity of a neuron transiently increases its firing threshold (see *Figure 1A* for an illustration). Experimental data from the Allen Institute (*Allen Institute, 2018b*) show that a substantial fraction of excitatory neurons of the neocortex, ranging from 20% in mouse visual cortex to 40% in the human frontal lobe, exhibit SFA (see *Appendix 1—figure 8*). Although a rigorous survey of time constants of SFA is still missing, the available experimental data show that SFA does produce history dependence of neural firing on the time scale of seconds, in fact, up to 20 s according to *Pozzorini et al., 2013*, *Pozzorini et al., 2015*. The biophysical mechanisms behind SFA include inactivation of depolarizing currents and the activity-dependent activation of slow hyperpolarizing or shunting currents (*Gutkin and Zeldenrust, 2014*; *Benda and Herz, 2003*).

SFA is an attractive feature from the perspective of the metabolic cost of neural coding and computation since it reduces firing activity (*Gutierrez and Denève, 2019*). But this increased metabolic efficiency comes at the cost of making spike codes for sensory stimuli history-dependent. Hence, it



**Figure 1.** Experimental data on neurons with spike frequency adaptation (SFA) and a simple model for SFA. (**A**) The response to a 1 s long step current is displayed for three sample neurons from the Allen brain cell database (***Allen Institute, 2018b***). The cell id and sweep number identify the exact cell recording in the Allen brain cell database. (**B**) The response of a simple leaky integrate-and-fire (LIF) neuron model with SFA to the 1-s-long step current. Neuron parameters used: top row $\beta = 0.5\,\text{mV}$, $\tau_a = 1\,\text{s}$, $I_{input} = 0.024\,\text{A}$; middle row $\beta = 1\,\text{mV}$, $\tau_a = 1\,\text{s}$, $I_{input} = 0.024\,\text{A}$; bottom row $\beta = 1\,\text{mV}$, $\tau_a = 300\,\text{ms}$, $I_{input} = 0.022\,\text{A}$. (**C**) Symbolic architecture of recurrent spiking neural network (SNN) consisting of LIF neurons with and without SFA. (**D**) Minimal SNN architecture for solving simple instances of STORE-RECALL tasks that we used to illustrate the negative imprinting principle. It consists of four subpopulations of input neurons and two LIF neurons with SFA, labeled NR and NL, that project to two output neurons (of which the stronger firing one provides the answer). (**E**) Sample trial of the network from (**D**) for two instances of the STORE-RECALL task. The input 'Right' is routed to the neuron NL, which fires strongly during the first STORE signal (indicated by a yellow shading of the time segment), that causes its firing threshold (shown at the bottom in blue) to strongly increase. The subsequent RECALL signal (green shading) excites both NL and NR, but NL fires less, that is, the storing of the working memory content 'Right' has left a 'negative imprint' on its

*Figure 1 continued on next page*

*Figure 1 continued*

excitability. Hence, NR fires stronger during recall, thereby triggering the answer 'Right' in the readout. After a longer pause, which allows the firing thresholds of NR and NL to reset, a trial is shown where the value 'Left' is stored and recalled.

becomes harder to decode information from spikes if neurons with SFA are involved (*Weber and Fairhall, 2019*; *Weber et al., 2019*). This problem appears already for very simple input streams, where the same stimulus is presented repeatedly, since each presentation is likely to create a somewhat different neural code in the network. However, it has recently been shown that a careful network construction can ensure that stable neural codes emerge on the network level (*Gutierrez and Denève, 2019*).

A number of potential computational advantages of neurons with SFA have already been identified, such as cellular short-term memory (*Marder et al., 1996*; *Turrigiano et al., 1996*), enhancement of sensitivity to synchronous input and desirable modifications of the frequency response curve (*Benda et al., 2010*; *Ermentrout, 1998*; *Wang, 1998*; *Gutkin and Zeldenrust, 2014*). On the network level, SFA may enhance rhythms and support Bayesian inference (*Kilpatrick and Ermentrout, 2011*; *Deneve, 2008*). The contribution of SFA to temporal computing capabilities of recurrent spiking neural networks (SNNs) had first been examined in *Bellec et al., 2018a*, and its role for language processing in feedforward networks was subsequently examined in *Fitz et al., 2020*.

Here we are taking a closer look at enhanced temporal computing capabilities of SNNs that are enabled through SFA and also compare the computational benefit of SFA with that of previously considered slower dynamic processes in SNNs: short-term synaptic plasticity. Most experimental analyses of temporal computing capabilities of biological neural networks have focused on arguably the simplest type of temporal computing, where the response to a stimulus has to be given after a delay. In other words, information about the preceding stimulus has to be kept in a working memory. We start with this simple task (STORE-RECALL task) since this task makes the analysis of neural coding especially transparent. We use it here to demonstrate a novel principle that is used by neurons with SFA to implement working memory – 'the negative imprinting principle.' That is, firing of a neuron leaves a negative imprint of its activity because its excitability is reduced due to SFA. Such negative imprinting has previously been utilized in *Gutierrez and Denève, 2019*. We then show that the working memory capability of SNNs with SFA scales up to much more demanding and ecologically more realistic working memory tasks, where not just a single but numerous features, for example, features that characterize a previously encountered image, movie scene, or sentence, have to be stored simultaneously.

However, these working memory tasks capture just a small fragment of temporal computing capabilities of brains. Substantially more common and more difficult are tasks where information is temporally dispersed over a continuous input stream, say in a sentence or a video clip, and has to be integrated over time in order to solve a task. This requires that the information that is stored in working memory has to be continuously updated. We tested temporal computing capabilities of SNNs with SFA for two types of such tasks. First, we consider standard benchmark tasks for temporal computing capabilities: keyword spotting, time-series classification (sequential MNIST), and delayed XOR task. Then we consider two tasks that are arguably at the heart of higher-level cognitive brain processing (*Lashley, 1951*): processing and manipulations of sequences of symbols according to dynamic rules. We also analyze the neural codes that emerge in SNNs with SFA for such tasks and compare them with neural codes for corresponding tasks in the brain (*Barone and Joseph, 1989*; *Liu et al., 2019*; *Carpenter et al., 2018*). Since our focus is on computing, rather than learning capabilities, we use a powerful tool for optimizing network parameters for task performance: backpropagation through time (BPTT). While this method is not assumed to be biologically realistic, it has recently been shown that almost the same task performance can, in general, be achieved for SNNs – with and without SFA – through training with a biologically more realistic learning method: *e-prop* (*Bellec et al., 2020*). We demonstrate this here for the case of the 12AX task.

## Results

### Network model

We employ a standard simple model for neurons with SFA, the generalized leaky integrate-and-fire (LIF) model $GLIF_2$ from **Teeter et al., 2018**; **Allen Institute, 2018a**. A practical advantage of this simple model is that it can be efficiently simulated and that it is amenable to gradient descent training (**Bellec et al., 2018a**). It is based on a standard LIF neuron model. In a LIF neuron, inputs are temporally integrated, giving rise to its membrane potential. The neuron produces a spike when its membrane potential is above a threshold $v_{th}$. After the spike, the membrane potential is reset and the neuron enters a refractory period during which it cannot spike again. The precise dynamics of the LIF model is given in **Equation (2)** in Materials and methods. The $GLIF_2$ model extends the LIF model by adding to the membrane voltage a second hidden variable, a variable component $a(t)$ of the firing threshold $A(t)$ that increases by a fixed amount after each spike $z(t)$ of the neuron, and then decays exponentially back to 0 (see **Figure 1B, E**). This variable threshold models the inactivation of voltage-dependent sodium channels in a qualitative manner. We write $z_j(t)$ for the spike output of neuron $j$, which switches from 0 to 1 at time $t$ when the neuron fires at time $t$, and otherwise has value 0. With this notation, one can define the SFA model by the equations

$$A_j(t) = v_{\text{th}} + \beta a_j(t),$$
$$a_j(t+1) = \rho_j a_j(t) + (1 - \rho_j) z_j(t),$$

(1)

where $v_{th}$ is the constant baseline of the firing threshold $A_j(t)$, and $\beta > 0$ scales the amplitude of the activity-dependent component. The parameter $\rho_j = \exp\left(\frac{-1}{\tau_{a,j}}\right)$ controls the speed by which $a_j(t)$ decays back to 0, where $\tau_{a,j}$ is the adaptation time constant of neuron $j$. For simplicity, we used a discrete-time model with a time step of $\delta t = 1$ ms (see Materials and methods for further details). We will in the following also refer to this model as 'LIF with SFA.' Consistent with the experimental data (**Allen Institute, 2018a**), we consider recurrent networks of LIF neurons, SNNs, of which some fraction is equipped with SFA. It turns out that the precise fraction of neurons with SFA does not matter for most tasks, especially if it stays somewhere in the biological range of 20–40%. We usually consider for simplicity fully connected recurrent networks, but most tasks can also be solved with sparser connectivity. Neurons in the recurrent network project to readout neurons, which produce the output of the network (see **Figure 1C**). The final output was either the maximum value after applying the softmax or thresholded values after applying the sigmoid on each readout neuron.

In order to analyze the potential contribution of SFA to temporal computing capabilities of SNNs, we optimized the weights of the SNN for each task. We used for this stochastic gradient descent in the form of BPTT (**Mozer, 1989**; **Robinson and Fallside, 1987**; **Werbos, 1988**), which is, to the best of our knowledge, the best-performing optimization method. Although this method performs best for differentiable neural network models, it turns out that the non-differentiable output of a spiking neuron can be overcome quite well with the help of a suitably scaled pseudo-derivative (**Bellec et al., 2018a**). In general, similar task performance can also be achieved with a biologically plausible learning method for SNNs e-prop (**Bellec et al., 2020**). Although computing rather than learning capabilities are in the focus of this paper, we demonstrate for one of the most demanding tasks that we consider, 12AX task, that almost the same task performance as with BPTT can be achieved with e-prop.

### SFA provides working memory simultaneously for many pieces of information and yields powerful generalization capability

To elucidate the mechanism by which SFA supports temporal computing capabilities of SNNs, we first consider classical working memory tasks, where information just has to be temporally stored by the neural network, without the need for frequent updates of this working memory during an instance of the task.

#### Negative imprinting principle

To demonstrate how neurons with SFA can contribute to solving working memory tasks, we first consider the standard case where just a single value, for example, the position left or right of a prior

stimulus, has to be stored during a delay. The simple network shown in *Figure 1D*, consisting of two neurons with SFA (NL and NR), can already solve this task if there are long gaps between different instances of the task. We assume that these two neurons receive spike inputs from four populations of neurons. Two of them encode the value that is to be stored, and the other two convey the commands STORE and RECALL through high-firing activity in these populations of input neurons (see *Figure 1E* for an illustration). The neuron NL (NR) fires when the population that encodes the STORE command fires (yellow shading in *Figure 1D*) and simultaneously the input population for value 'Right' ('Left') is active. Furthermore, we assume that the input population that encodes the RECALL command (green shading in *Figure 1E*) causes both NL and NR to fire. However, the firing threshold of that one of them that had fired during the preceding STORE command is higher (see the blue threshold in the left half and the red threshold in the right half of *Figure 1E*), causing a weaker response to this RECALL command. Hence, if the spikes of NL and NR are each routed to one of the two neurons in a subsequent winner-take-all (WTA) circuit, the resulting winner encodes the value that has been stored during the preceding STORE command. The time courses of the firing thresholds of neurons NL and NR in *Figure 1E* clearly indicate the negative imprinting principle that underlies this working memory mechanism: the neuron that fires less during RECALL was the one that had responded the strongest during the preceding STORE phase, and this firing left a stronger 'negative imprint' on this neuron. Note that this hand-constructed circuit does not work for large ranges of time differences between STORE and RECALL, and more neurons with SFA are needed to solve the subsequently discussed full versions of the task.

## Scaling the negative imprinting principle up to more realistic working memory tasks

We wondered whether SNNs with SFA can also solve more realistic working memory tasks, where not just a single bit, but a higher-level code of a preceding image, sentence, or movie clip needs to be stored. Obviously, brains are able to do that, but this has rarely been addressed in models. In addition, brains are generally exposed to ongoing input streams also during the delay between STORE and RECALL, but need to ignore these irrelevant input segments. Both of these more demanding aspects are present in the more demanding version of the STORE-RECALL task that is considered in *Figure 2A*. Here the values of 20, instead of just 1, input bits need to be stored during a STORE command and recalled during a RECALL command. More precisely, a 20-dimensional stream of input bits is given to the network, whose values during each 200 ms time segment are visualized as 4 × 5 image in the top row of *Figure 2A*. Occasionally, a pattern in the input stream is marked as being salient through simultaneous activation of a STORE command in a separate input channel, corresponding, for example, to an attentional signal from a higher brain area (see yellow shading in *Figure 2A*). The task is to reproduce during a RECALL command the pattern that had been presented during the most recent STORE command. Delays between STORE and RECALL ranged from 200 to 1600 ms. 20 binary values were simultaneously extracted as network outputs during RECALL by thresholding the output values of 20 linear readout neurons. We found that an SNN consisting of 500 neurons with SFA, whose adaptive firing threshold had a time constant of $\tau_a = 800$ ms, was able to solve this task with an accuracy above 99% and average firing activity of $13.90 \pm 8.76$ Hz (mean $\pm$ standard deviation). SFA was essential for this behavior because the recall performance of a recurrent network of LIF neurons without SFA, trained in exactly the same way, stayed at chance level (see Materials and methods). In *Figure 2A*, one sees that those neurons with SFA that fired stronger during STORE fire less during the subsequent RECALL, indicating a use of the negative imprinting principle also for this substantially more complex working memory task.

Interestingly, this type of working memory in an SNN with SFA shares an important feature with the activity-silent form of working memory in the human brain that had been examined in the experiments of *Wolff et al., 2017*. It had been shown there that the representation of working memory content changes significantly between memory encoding and subsequent network reactivation during the delay by an 'impulse stimulus': a classifier trained on the network activity during encoding was not able to classify the memory content during a network reactivation in the delay, and vice versa. Obviously, this experimental result from the human brain is consistent with the negative imprinting principle. We also tested directly whether the experimentally observed change of neural codes in the human brain also occurs in our model. We trained a classifier for decoding the content



**Figure 2.** 20-dimensional STORE-RECALL and sMNIST task. (**A**) Sample trial of the 20-dimensional STORE-RECALL task where a trained spiking neural network (SNN) of leaky integrate-and-fire (LIF) neurons with spike frequency adaptation (SFA) correctly stores (yellow shading) and recalls (green shading) a pattern. (**B, C**) Test accuracy comparison of recurrent SNNs with different slow mechanisms: dual version of SFA where the threshold is decreased and causes enhanced excitability (ELIF), predominantly depressing (STP-D) and predominantly facilitating short-term plasticity (STP-F). (**B**) Test set accuracy of five variants of the SNN model on the one-dimensional STORE-RECALL task. Bars represent the mean accuracy of 10 runs with different network initializations. (**C**) Test set accuracy of the same five variants of the SNN model for the sMNIST time-series classification task. Bars represent the mean accuracy of four runs with different network initializations. Error bars in (**B**) and (**C**) indicate standard deviation.

of working memory during STORE and found that this classifier was not able to decode this content during RECALL, and vice versa (see Materials and methods). Hence, our model is in this regard consistent with the experimental data of *Wolff et al., 2017*.

We also found that it was not possible to decode the stored information from the firing activity between STORE and RECALL, as one would expect if the network would store the information through persistent firing. Actually, the firing activity was quite low during this time period. Hence, this demo shows that SNNs with SFA have, in addition to persistent firing, a quite different method for transient storage of information.

## Generalization of SFA-enhanced temporal computations to unseen inputs

In contrast to the brain, many neural network models for working memory can store only information on which they have been trained. In fact, this tends to be unavoidable if a model can only store a single bit. In contrast, the human brain is also able to retain new information in its working memory. The SNN with SFA that we used for the 20-dimensional working memory task also had this capability. It achieved a performance of 99.09%, that is, 99.09% of the stored 20-dimensional bit vectors were accurately reproduced during recall, on bit vectors that had never occurred during training. In fact, we made sure that all bit vectors that had to be stored during testing had a Hamming distance of at least five bits to all bit vectors used during training. A sample segment of a test trial is shown in *Figure 2A*, with the activity of input neurons at the top and the activation of readout neurons at the bottom.

## No precise alignment between time constants of SFA and working memory duration is needed

Experimental data from the Allen Institute database suggest that different neurons exhibit a diversity of SFA properties. We show that correspondingly a diversity of time constants of SFA in different neurons provides high performance for temporal computing. We consider for simplicity the one-dimensional version of the task of *Figure 2A*, where just a single bit needs to be stored in working memory between STORE and RECALL commands. The expected delay between STORE and RECALL (see the header row of *Table 1*) scales the working memory time span that is required to solve this task. Five fixed time constants were tested for SFA ($\tau_a = 200$ ms, 2 s, 4 s, 8 s, see top five rows of *Table 1*). Also, a power-law distribution of these time constants, as well as a uniform distribution, was considered (see last two rows of *Table 1*). One sees that the resulting diversity of time constants for SFA yields about the same performance as a fixed choice of the time constant that is aligned

**Table 1.** Recall accuracy (in %) of spiking neural network (SNN) models with different time constants of spike frequency adaptation (SFA) (rows) for variants of the STORE-RECALL task with different required memory time spans (columns).

Good task performance does not require good alignment of SFA time constants with the required time span for working memory. An SNN consisting of 60 leaky integrate-and-fire (LIF) neurons with SFA was trained for many different choices of SFA time constants for variations of the one-dimensional STORE-RECALL task with different required time spans for working memory. A network of 60 LIF neurons without SFA trained under the same parameters did not improve beyond chance level (~50% accuracy), except for the task instance with an expected delay of 200 ms where the LIF network reached 96.7% accuracy (see top row).

| Expected delay between STORE and RECALL | 200 ms | 2 s | 4 s | 8 s | 16 s |
|---|---|---|---|---|---|
| Without SFA ($\tau_a = 0$ ms) | 96.7 | 51 | 50 | 49 | 51 |
| $\tau_a = 200$ ms | 99.92 | 73.6 | 58 | 51 | 51 |
| $\tau_a = 2$ s | 99.0 | 99.6 | 98.8 | 92.2 | 75.2 |
| $\tau_a = 4$ s | 99.1 | 99.7 | 99.7 | 97.8 | 90.5 |
| $\tau_a = 8$ s | 99.6 | 99.8 | 99.7 | 97.7 | 97.1 |
| $\tau_a$ power-law dist. in [0, 8] s | 99.6 | 99.7 | 98.4 | 96.3 | 83.6 |
| $\tau_a$ uniform dist. in [0, 8] s | 96.2 | 99.9 | 98.6 | 92.1 | 92.6 |

with the required memory span of the task. However, a much larger time constant (see the row with $\tau_a = 8$ s in the column with an expected memory span of 200 ms or 2 s for the task) or a substantially smaller time constant (see the row with $\tau_a = 2$ s in the column with an expected memory span of 8 s) tends to work well.

## SFA improves the performance of SNNs for common benchmark tasks that require nonlinear computational operations on temporally dispersed information

We now turn to more demanding temporal computing tasks, where temporally dispersed information not only needs to be stored, but continuously updated in view of new information. We start out in this section with three frequently considered benchmark tasks of this type: sequential MNIST, Google Speech Commands, and delayed XOR.

Sequential MNIST (sMNIST) is a standard benchmark task for time-series classification. In this variant of the well-known handwritten digit recognition dataset MNIST, the pixel values of each sample of a handwritten digit are presented to the network in the form of a time series, one pixel in each ms, as they arise from a row-wise scanning pattern of the handwritten digit. This task also requires very good generalization capability since the resulting sequence of pixel values for different handwriting styles of the same digit may be very different, and the network is tested on samples that had never before been presented to the network.

An SNN with SFA was able to solve this task with a test accuracy of 93.7%, whereas an SNN without SFA only reached an accuracy of 51.8%. We refer to Section 2 of Appendix 1 for further details.

We also compared the performance of SNNs with and without SFA on the keyword-spotting task Google Speech Commands Dataset (*Warden, 2018*) (v0.02). To solve this task, the network needs to correctly classify audio recordings of silence, spoken unknown words, and utterings of 1 of 10 keywords by different speakers. On this task, the performance of SNNs increases with the inclusion of SFA (from 89.04% to 91.21%) and approaches the state-of-the-art artificial recurrent model (93.18%) (see Section 3 of Appendix 1 and *Appendix 1—table 1*).

Finally, we tested the performance of SNNs with SFA on the delayed-memory XOR task, a task which had previously already been used as benchmark tasks for SNNs in *Huh and Sejnowski, 2018*. In this task, the network is required to compute the exclusive-or operation on a time series of binary input pulses and provide the answer when prompted by a go-cue signal. Across 10 different runs, an SNN with SFA solved the task with $95.19 \pm 0.014\%$ accuracy, whereas the SNN without SFA just achieved $61.30 \pm 0.029\%$ (see Section 4 of Appendix 1 and *Appendix 1—figure 3*).

The good performance of SNNs with SFA on all three tasks demonstrates that SFA provides computational benefits to SNNs also for substantially more demanding temporal computing tasks in comparison with standard working memory tasks. Before we turn to further temporal computing tasks that are of particular interest from the perspective of neuroscience and cognitive science, we first analyze the contribution of other slow mechanisms in biological neurons and synapses on the basic working memory task and on sMNIST.

## Comparing the contribution of SFA to temporal computing with that of other slow processes in neurons and synapses

Facilitating short-term plasticity (STP-F) and depressing short-term plasticity (STP-D) are the most frequently discussed slower dynamic processes in biological synapses. STP-F of synapses, also referred to as paired-pulse facilitation, increases the amplitudes of postsynaptic potentials for the later spikes in a spike train. Whereas synaptic connections between pyramidal cells in the neocortex are usually depressing (*Markram et al., 2015*), it was shown in *Wang et al., 2006* that there are facilitating synaptic connections between pyramidal cells in the medial prefrontal cortex of rodents, with a mean time constant of 507 ms (standard deviation 37 ms) for facilitation. It was shown in *Mongillo et al., 2008* that if one triples the experimentally found mean time constant for facilitation, then this mechanism supports basic working memory tasks.

STP-D of synapses, also referred to as paired-pulse depression, reduces the amplitude of postsynaptic potentials for later spikes in a spike train. The impact of this mechanism on simple temporal computing tasks had been examined in a number of publications (*Maass et al., 2002*; *Buonomano and Maass, 2009*; *Masse et al., 2019*; *Hu et al., 2020*).

In addition, we consider a dual version of SFA: a neuron model where each firing of the neuron causes its firing threshold to decrease – rather than increase as in SFA – which then returns exponentially to its resting value. We call this neuron model the enhanced-excitability LIF (ELIF) model. Such neural mechanism had been considered, for example, in *Fransén et al., 2006*. Note that a transient increase in the excitability of a neuron can also be caused by depolarization-mediated suppression of inhibition, a mechanism that has been observed in many brain areas (*Kullmann et al., 2012*). The dynamics of the salient hidden variables in all three models are described in Materials and methods and illustrated in *Appendix 1—figure 2*.

We tested the resulting five different types of SNNs, each consisting of 60 neurons, first on the simple 1D working memory task. The results in *Figure 2B* show that SNNs with SFA provide by far the best performance on this task.

*Figure 2C* shows that for sMNIST both SNNs with SFA and SNNs with STP-D achieve high performance. Surprisingly, the performance of SNNs with facilitating synapses is much worse, both for sMNIST and for the working memory task.

## SFA supports demanding cognitive computations on sequences with dynamic rules

Complex cognitive tasks often contain a significant temporal processing component, including the requirement to flexibly incorporate task context and rule changes. To test whether SFA can support such cognitive processing, we consider the 12AX task (*Frank et al., 2001*). This task is an extension of the A-X version of the continuous performance task (CPT-AX), which has been extensively studied in humans (*Barch et al., 2009*). It tests the ability of subjects to apply dynamic rules when detecting specific subsequences in a long sequence of symbols while ignoring irrelevant inputs (*O'Reilly and Frank, 2006*; *MacDonald, 2008*). It also probes the capability to maintain and update a hierarchical working memory since the currently active rule, that is, the context, stays valid for a longer period of time and governs what other symbols should be stored in working memory.

More precisely, in the 12AX task, the subject is shown a sequence of symbols from the set 1, 2, A, X, B, Y, C, Z. After processing any symbol in the sequence, the network should output 'R' if this symbol terminates a context-dependent target sequence and 'L' otherwise. The current target sequence depends on the current context, which is defined through the symbols '1' and '2.' If the most recently received digit was a '1', the subject should output 'R' only when it encounters a symbol 'X' that terminates a subsequence A...X. This occurs, for example, for the seventh symbol in the trial shown in *Figure 3*. In case that the most recent input digit was a '2', the subject should instead respond 'R' only after the symbol 'Y' in a subsequent subsequence B...Y (see the 20th symbol in *Figure 3*). In addition, the processed sequence contains letters 'C' and 'Z' that are irrelevant and serve as distractors. This task requires a hierarchical working memory because the most recently occurring digit determines whether subsequent occurrences of 'A' or 'B' should be placed into working memory. Note also that neither the content of the higher-level working memory, that is, the digit, nor the content of the lower-level working memory, that is, the letter A or B, are simply recalled. Instead, they affect the target outputs of the network in a more indirect way. Furthermore, the higher-level processing rule affects what is to be remembered at the lower level.

A simpler version of this task, where X and Y were relevant only if they directly followed A or B, respectively, and where fewer irrelevant letters occurred in the input, was solved in *O'Reilly and Frank, 2006*; *Martinolli et al., 2018*; *Kruijne et al., 2020* through biologically inspired artificial neural network models that were endowed with special working memory modules. Note that for this simpler version no lower-order working memory is needed because one just has to wait for an immediate transition from A to X in the input sequence or for an immediate transition from B to Y. But neither the simpler nor the more complex version, which is considered here, of the 12AX task has previously been solved by a network of spiking neurons.

In the version of the task that we consider, the distractor symbols between relevant symbols occur rather frequently. Hence, robust maintenance of relevant symbols in the hierarchical working memory becomes crucial because time spans between relevant symbols become longer, and hence the task is more demanding – especially for a neural network implementation.

Overall, the network received during each trial (episode) sequences of 90 symbols from the set {1, 2, A, B, C, X, Y, Z}, with repetitions as described in Materials and methods. See the top of *Figure 3* for an example (the context-relevant symbols are marked in bold for visual ease).

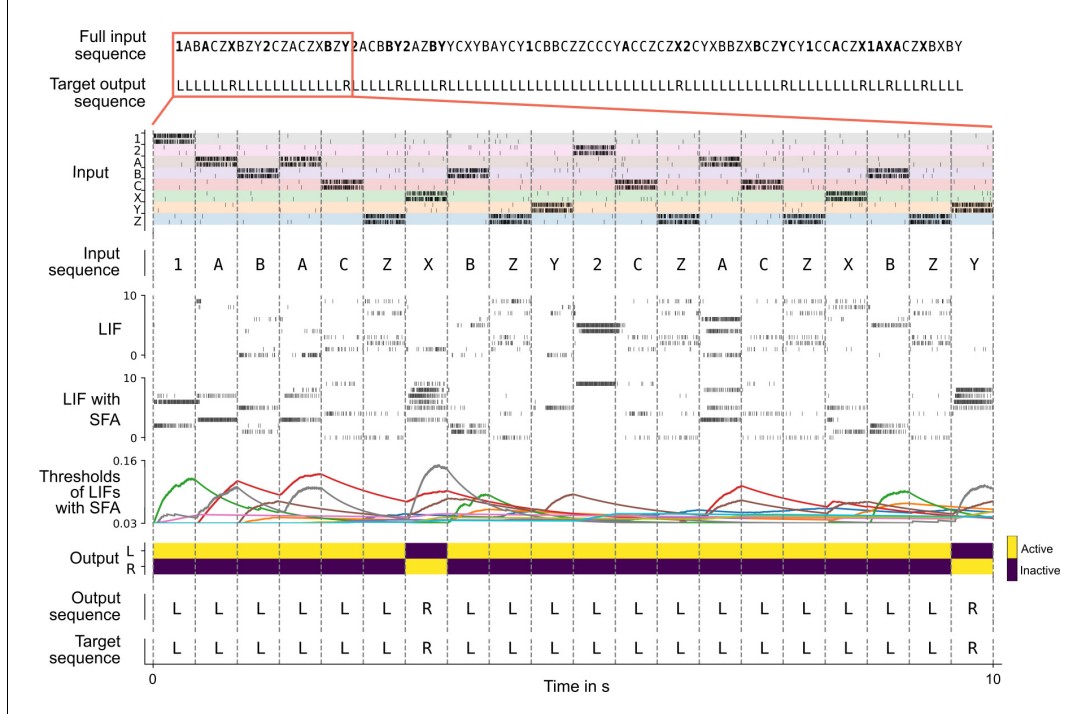

**Figure 3.** Solving the 12AX task by a network of spiking neurons with spike frequency adaptation (SFA). A sample trial of the trained network is shown. From top to bottom: full input and target output sequence for a trial, consisting of 90 symbols each, blow-up for a subsequence – spiking input for the subsequence, the considered subsequence, firing activity of 10 sample leaky integrate-and-fire (LIF) neurons without and 10 sample LIF neurons with SFA from the network, time course of the firing thresholds of these neurons with SFA, output activation of the two readout neurons, the resulting sequence of output symbols which the network produced, and the target output sequence.

We show in *Figure 3* that a generic SNN with SFA can solve this quite demanding version of the 12AX task. The network consisted of 200 recurrently connected spiking neurons (100 with and 100 without SFA), with all-to-all connections between them. After training, for new symbol sequences that had never occurred during training, the network produced an output string with all correct symbols in 97.79% of episodes.

The average firing activity of LIF neurons with SFA and LIF neurons without SFA was $(12.37 \pm 2.90)$ Hz and $(10.65 \pm 1.63)$ Hz (mean $\pm$ standard deviation), respectively (the average was calculated over 2000 test episodes for one random initialization of the network). Hence, the network operated in a physiologically meaningful regime.

These results were obtained after optimizing synaptic weights via BPTT. However, training with a recently published biologically plausible learning method called *random e-prop* (*Bellec et al., 2020*) produced a similar performance of 92.89% (averaged over five different network initializations).

We next asked how the fraction of neurons with SFA affects SNN performance in the case of the usual parameter optimization via BPTT. When all, rather than just half of the 200, LIF neurons were endowed with SFA, a much lower accuracy of just 72.01% was achieved. On the other hand, if just 10% of the neurons had SFA, a performance of 95.39% was achieved. In contrast, a recurrent SNN with the same architecture but no neurons with SFA only achieved a performance of 0.39% (each success rate was averaged over five network initializations). Hence, a few neurons with SFA suffice for good performance, and it is important to also have neurons without SFA for this task.

Neuronal networks in the brain are subject to various sources of noise. A highly optimized SNN model with sparse firing activity might utilize brittle spike-time correlations. Such a network would therefore be highly susceptible to noise. To test whether this was the case in our model, we tested how the performance of the above network changed when various levels of noise were added to all network neurons during testing. We found that although the spike responses of the neurons become quite different, see *Appendix 1—figures 5* and *6*, the performance of the SNN model is little

affected by low noise and decays gracefully for higher levels of noise. For details, see Section 5 of Appendix 1.

Surprisingly, it was not necessary to create a special network architecture for the two levels of working memory that our more complex version of the 12AX task requires: a near perfectly performing network emerged from training a generic fully connected SNN with SFA.

## SFA enables SNNs to carry out complex operations on sequences of symbols

Learning to carry out operations on sequences of symbols in such a way that they generalize to new sequences is a fundamental capability of the human brain, but a generic difficulty for neural networks (*Marcus, 2003*). Not only humans but also non-human primates are able to carry out operations on sequences of items, and numerous neural recordings starting with (*Barone and Joseph, 1989*) up to recent results such as (*Carpenter et al., 2018*; *Liu et al., 2019*) provide information about the neural codes for sequences that accompany such operations in the brain. The fundamental question of how serial order of items is encoded in working memory emerges from the more basic question of how the serial position of an item is combined with the content information about its identity (*Lashley, 1951*). The experimental data both of *Barone and Joseph, 1989* and *Liu et al., 2019* suggest that the brain uses a factorial code where position and identity of an item in a sequence are encoded separately by some neurons, thereby facilitating flexible generalization of learned experience to new sequences.

We show here that SNNs with SFA can be trained to carry out complex operations on sequences, are able to generalize such capabilities to new sequences, and produce spiking activity and neural codes that can be compared with neural recordings from the brain. In particular, they also produce factorial codes, where separate neurons encode the position and identity of a symbol in a sequence. One basic operation on sequences of symbols is remembering and reproducing a given sequence (*Liu et al., 2019*). This task had been proposed by *Marcus, 2003* to be a symbolic computation task that is fundamental for symbol processing capabilities of the human brain. But non-human primates can also learn simpler versions of this task, and hence it was possible to analyze how neurons in the brain encode the position and identity of symbols in a sequence (*Barone and Joseph, 1989*; *Carpenter et al., 2018*). Humans can also reverse sequences, a task that is more difficult for artificial networks to solve (*Marcus, 2003*; *Liu et al., 2019*). We show that an SNN with SFA can carry out both of these operations and is able to apply them to new sequences of symbols that did not occur during the training of the network.

We trained an SNN consisting of 320 recurrently connected LIF neurons (192 with and 128 without SFA) to carry out these two operations on sequences of 5 symbols from a repertoire of 31 symbols. Once trained, the SNN with SFA could duplicate and reverse sequences that it had not seen previously, with a success rate of 95.88% (average over five different network initializations). The 'success rate' was defined as the fraction of test episodes (trials) where the full output sequence was generated correctly. Sample episodes of the trained SNN are shown in *Figure 4A*, and a zoom-in of the same spike rasters is provided in *Appendix 1—figure 7*. For comparison, we also trained a LIF network without SFA in exactly the same way with the same number of neurons. It achieved a performance of 0.0%.

The average firing activity of LIF neurons without SFA and LIF neurons with SFA was $(19.88 \pm 2.68)$ Hz, and $(21.51 \pm 2.95)$ Hz (mean $\pm$ standard deviation), respectively. The average was calculated over $50,000$ test episodes for one random initialization of the network.

## A diversity of neural codes emerge in SNNs with SFA trained to carry out operations on sequences

Emergent coding properties of neurons in the SNN are analyzed in *Figure 4B–D*, and two sample neurons are shown in *Figure 4E, F*. Neurons are sorted in *Figure 4B, C* according to the time of their peak activity (averaged over 1000 episodes), like in *Harvey et al., 2012*. The neurons have learned to abstract the overall timing of the tasks (*Figure 4B*). A number of network neurons (about one-third) participate in sequential firing activity independent of the type of task and the symbols involved (see the lower part of *Figure 4B* and the trace for the average activity of neurons left of the marker for the start of duplication or reversal). This kind of activity is reminiscent of the neural

**Figure 4.** Analysis of a spiking neural network (SNN) with spike frequency adaptation (SFA) trained to carry out operations on sequences. (**A**) Two sample episodes where the network carried out sequence duplication (left) and reversal (right). Top to bottom: spike inputs to the network (subset), sequence of symbols they encode, spike activity of 10 sample leaky integrate-and-fire (LIF) neurons (without and with SFA) in the SNN, firing threshold dynamics for these 10 LIF neurons with SFA, activation of linear readout neurons, output sequence produced by applying argmax to them, and target

*Figure 4 continued on next page*

*Figure 4 continued*

output sequence. (**B–F**) Emergent neural coding of 279 neurons in the SNN (after removal of neurons detected as outliers) and peri-condition time histogram (PCTH) plots of two sample neurons. Neurons are sorted by time of peak activity. (**B**) A substantial number of neurons were sensitive to the overall timing of the tasks, especially for the second half of trials when the output sequence is produced. (**C**) Neurons separately sorted for duplication episodes (top row) and reversal episodes (bottom row). Many neurons responded to input symbols according to their serial position, but differently for different tasks. (**D**) Histogram of neurons categorized according to conditions with statistically significant effect (three-way ANOVA). Firing activity of a sample neuron that fired primarily when (**E**) the symbol 'g' was to be written at the beginning of the output sequence. The activity of this neuron depended on the task context during the input period; (**F**) the symbol 'C' occurred in position 5 in the input, irrespective of the task context.

The online version of this article includes the following source data for figure 4:

**Source data 1.** Raw data to generate Figure 4DEF.

activity relative to the start of a trial that was recorded in rodents after they had learned to solve tasks that had a similar duration (*Tsao et al., 2018*).

The time of peak activity of other neurons in *Figure 4B* depended on the task and the concrete content, indicated by a weak activation during the loading of the sequence (left of the marker), but stronger activation after the start of duplication or reversal (right of the marker). The dependence on the concrete content and task is shown in *Figure 4C*. Interestingly, these neurons change their activation order already during the loading of the input sequence in dependence of the task (duplication or reversal). Using three-way ANOVA, we were able to categorize each neuron as selective to a specific condition (symbol identity, serial position in the sequence, and type of task) or a nonlinear combination of conditions based on the effect size $\omega^2$. Each neuron could belong to more than one category if the effect size was above the threshold of 0.14 (as suggested by *Field, 2013*). Similar to recordings from the brain (*Carpenter et al., 2018*), a diversity of neural codes emerged that encode one variable or a combination of variables. In other words, a large fraction of neurons encoded a nonlinear combinations of all three variables (see *Figure 4D*). Peri-condition time histogram (PCTH) plots of two sample neurons are shown in *Figure 4E, F*: one neuron is selective to symbol 'g' but at different positions depending on task context; the other neuron is selective to symbol 'C' occurring at position 5 in the input, independent of task context. Thus one sees that a realization of this task by an SNN, which was previously not available, provides rich opportunities for a comparison of emergent spike codes in the model and neuronal recordings from the brain. For more details, see the last section of Materials and methods.

## Discussion

Brains are able to carry out complex computations on temporally dispersed information, for example, on visual inputs streams, or on sequences of words or symbols. We have addressed the question how the computational machinery of the brain, recurrently connected networks of spiking neurons, can accomplish that.

The simplest type of temporal computing task just requires to hold one item, which typically can be characterized by a single bit, during a delay in a working memory, until it is needed for a behavioral response. This can be modeled in neural networks by creating an attractor in the network dynamics that retains this bit of information through persistent firing during the delay. But this implementation is inherently brittle, especially for SNNs, and it is not clear whether it can be scaled up to more ecological working memory tasks where multiple features, for example, the main features that characterize an image or a story, are kept in working memory, even in the presence of a continuous stream of distracting network inputs. We have shown that SFA enables SNNs to solve this task, even for feature vector that the network had never encountered before (see *Figure 2A*).

This model for working memory shares many properties with how the human brain stores content that is not attended (*Wolff et al., 2017*):

1. The data of *Wolff et al., 2017* suggest that in order to understand such working memory mechanisms it is necessary to 'look beyond simple measures of neural activity and consider a richer diversity of neural states that underpin content-dependent behavior.' We propose that the current excitability of neurons with SFA is an example for such hidden neural state that is highly relevant in this context. This provides a concrete experimentally testable hypothesis.

2. They proposed more specifically that 'activity silent neural states are sufficient to bridge memory delays.' We have shown this in *Figure 2A* for a quite realistic working memory task, where a complex feature vector has to be kept in memory, also in the presence of continuous distractor inputs.

3. They found that an unspecific network input, corresponding to the activation of a population of input neurons for RECALL in our model, is able to recover in the human brain an item that has been stored in working memory, but that storing and recalling of an unattended item by 'pinging' the brain generates very different network activity. This is exactly what happens in our model. We have shown that a classifier that was trained to decode the stored item from the neural activity during encoding (STORE) was not able to decode the working memory content during RECALL, and vice versa. Furthermore, we have elucidated a particular neural coding principle, the negative imprinting principle, that is consistent with this effect (see the illustration in *Figure 1E*). An immediate experimentally testable consequence of the negative encoding principle is that the same network responds with reduced firing to repeated representations of an unattended item. This has in fact already been demonstrated for several brain areas, such as sensory cortices (*Kok and de Lange, 2015*) and perirhinal cortex (*Winters et al., 2008*).

4. They found that decoding of an unattended working memory item without 'pinging' the network 'dropped to chance relatively quickly after item presentation.' We found that also in our model the content of working memory could not be decoded during the delay between STORE and RECALL.

But obviously ecologically relevant temporal computing tasks that the brain routinely solves are much more complex and demanding than such standard working memory tasks. The 12AX is a prime example for such more demanding temporally computing task, where two different types of memories need to be continuously updated: memory about the currently valid rule and memory about data. The currently valid rule determines which item needs to be extracted from the continuous stream of input symbols and remembered: the symbol A or the symbol B. We have shown that SFA enables an SNN to solve this task, without requiring a specific architecture corresponding to the two types of working memory that it requires. This result suggests that it is also unlikely that such two-tiered architecture can be found in the brain, rather that both types of working memory are intertwined in the neural circuitry.

Our result suggests that most other temporal computing tasks that brains are able to solve can also be reproduced by such simple models. We have tested this hypothesis for another cognitively demanding task on temporally dispersed information that has been argued to represent an important 'atom of neural computation' in the brain, that is, an elementary reusable computing primitive on which the astounding cognitive capabilities of the human brain rely (*Marcus, 2003*; *Marcus et al., 2014*): the capability to reproduce or invert a given sequence of symbols, even if this sequence has never been encountered before. We have shown in *Figure 4* that SFA enables SNNs to solve this task. Since also monkeys can be trained to carry out simple operations on sequences of symbols, there are in this case experimental data available on neural codes that are used by the primate brain to encode serial order and identity of a sequence item. We found that, like in the brain, a diversity of neural codes emerge: neurons encode one or several of the relevant variables – symbol identity, serial position of a symbol, and type of task. Such comparison of neural coding properties of brains and neural network models is only possible if the model employs – like the brain – spiking neurons, and if the firing rates of these neurons remain in a physiological range of sparse activity, as the presented SNN models tend to provide. Hence, the capability to produce such brain-like computational capabilities in SNNs is likely to enhance the convergence of further biological experiments, models, and theory for uncovering the computational primitives of the primate brain.

Since there is a lack of further concrete benchmark tasks from neuroscience and cognitive science for temporal computing capabilities of brains, we have also tested the performance of SNNs with SFA on some benchmark tasks that are commonly used in neuromorphic engineering and AI, such as sequential MNIST and the Google Speech Commands Dataset. We found that SNNs with SFA can solve also these tasks very well, almost as well as the state-of-the-art models in machine learning and AI: artificial neural networks with special – unfortunately biologically implausible – units for temporal computing such as long short-term memory (LSTM) units.

Besides SFA, there are several other candidates for hidden states of biological neurons and synapses that may support brain computations on temporally dispersed information. We examined

three prominent candidates for other hidden states and analyzed how well they support these computations in comparison with SFA: depressing and facilitating short-term plasticity of synapses, as well as an activity-triggered increase in the excitability of neurons (ELIF neurons). We have shown in *Figure 2B* that these other types of hidden states provide lower performance than SFA for the simple working memory task. However, for a more demanding time-series classification task short-term depression of synapses provides about the same performance as SFA (see *Figure 2C*). An important contribution of depressing synapses for temporal computing has already previously been proposed (*Hu et al., 2020*). This is on first sight counter-intuitive, just as the fact that spike-triggered reduction rather than increase of neural excitability provides better support for temporal computing. But a closer look shows that it just requires a 'sign-inversion' of readout units to extract information from reduced firing activity. On the other hand, reduced excitability of neurons has the advantage that this hidden state is better protected against perturbations by ongoing network activity: a neuron that is in an adapted state where it is more reluctant to fire is likely to respond less to noise inputs, thereby protecting its hidden state from such noise. In contrast, a more excitable neuron is likely to respond also to weaker noise inputs, thereby diluting its hidden state. These observations go in a similar direction as the results of *Mongillo et al., 2018*; *Kim and Sejnowski, 2021*, which suggest that inhibition, rather than excitation, is critical for robust memory mechanisms in the volatile cortex.

Finally, it should be pointed out that there are numerous other hidden states in biological neurons and synapses that change on slower time scales. One prominent example are metabotropic glutamate receptors, which are present in a large fraction of synapses throughout the thalamus and cortex. Metabotropic receptors engage a complex molecular machinery inside the neuron, which integrates signals also over long time scales from seconds to hours and days (*Sherman, 2014*). However, at present, we are missing mathematical models for these processes, and hence it is hard to evaluate their contribution to temporal computing.

We have analyzed here the capabilities and limitations of various types of SNNs and have not addressed the question of how such capabilities could be induced in SNNs of the brain. Hence, we have used the most powerful optimization method for inducing computational capabilities in SNNs: a spike-oriented adaptation of BPTT. It was previously shown in *Bellec et al., 2020* that in general almost the same performance can be achieved in SNNs with SFA when BPTT is replaced by *e-prop*, a more biologically plausible network gradient descent algorithm. We have tested this also for the arguably most difficult temporal computing task that was examined in this paper, 12AX, and found that *e-prop* provides almost the same performance. However, temporal computing capabilities are likely to arise in brains through a combination of nature and nurture, and it remains to be examined to what extent the genetic code endows SNNs of the brain with temporal computing capabilities. In one of the first approaches for estimating the impact of genetically encoded connection probabilities on computational capabilities, it was already shown that connection probabilities can provide already some type of working memory without any need for learning (*Stöckl et al., 2021*), but this approach has not yet been applied to SNNs with SFA or other slowly changing hidden states.

Finally, our results raise the question whether the distribution of time constants of SFA in a cortical area is related to the intrinsic time scale of that cortical area, as measured, for example, via intrinsic fluctuations of spiking activity (*Murray et al., 2014*; *Wasmuht et al., 2018*). Unfortunately, we are lacking experimental data on time constants of SFA in different brain areas. We tested the relation between time constants of SFA and the intrinsic time scale of neurons according to *Wasmuht et al., 2018* for the case of the STORE-RECALL task (see Section 1 of Appendix 1 and *Appendix 1—figure 1*). We found that the time constants of neurons with SFA had little impact on their intrinsic time scale for this task, in particular much less than the network input. We have also shown in control experiments that the alignment between the time scale of SFA and the time scale of working memory duration can be rather loose. Even a random distribution of time constants for SFA works well.

Altogether, we have shown that SFA, a well-known feature of a substantial fraction of neurons in the neocortex, provides an important new facet for our understanding of computations in SNNs: it enables SNNs to integrate temporally dispersed information seamlessly into ongoing network computations. This paves the way for reaching a key goal of modeling: to combine detailed experimental data from neurophysiology on the level of neurons and synapses with the brain-like high computational performance of the network.

## Materials and methods

In this section, we first describe the details of the network models that we employ, and then we continue with the description of the training methods. After that, we give details about all the tasks and analyses performed.

### Network models

#### LIF neurons

A LIF neuron $j$ spikes as soon at its membrane potential $V_j(t)$ is above its threshold $v_{\text{th}}$. At each spike time $t$, the membrane potential $V_j(t)$ is reset by subtracting the threshold value $v_{\text{th}}$ and the neuron enters a strict refractory period for 3–5 ms (depending on the experiment) where it cannot spike again. Between spikes, the membrane voltage $V_j(t)$ is following the dynamics

$$\tau_m \dot{V}_j(t) = -V_j(t) + R_m I_j(t), \tag{2}$$

where $\tau_m$ is the membrane constant of neuron $j$, $R_m$ is the resistance of the cell membrane, and $I_j$ is the input current.

Our simulations were performed in discrete time with a time step $\delta t = 1$ ms. In discrete time, the input and output spike trains are modeled as binary sequences $x_i(t), z_j(t) \in \{0, \frac{1}{\delta t}\}$, respectively. Neuron $j$ emits a spike at time $t$ if it is currently not in a refractory period, and its membrane potential $V_j(t)$ is above its threshold. During the refractory period following a spike, $z_j(t)$ is fixed to 0. The neural dynamics in discrete time reads as follows:

$$V_j(t + \delta t) = \alpha V_j(t) + (1 - \alpha) R_m I_j(t) - v_{\text{th}} z_j(t) \delta t, \tag{3}$$

where $\alpha = \exp(-\frac{\delta t}{\tau_m})$, with $\tau_m$ being the membrane constant of the neuron $j$. The spike of neuron $j$ is defined by $z_j(t) = H\left(\frac{V_j(t) - v_{\text{th}}}{v_{\text{th}}}\right)\frac{1}{\delta t}$, with $H(x) = 0$ if $x < 0$ and 1 otherwise. The term $-v_{\text{th}} z_j(t) \delta t$ implements the reset of the membrane voltage after each spike.

In all simulations, the $R_m$ was set to $1\,\text{G}\Omega$. The input current $I_j(t)$ is defined as the weighted sum of spikes from external inputs and other neurons in the network:

$$I_j(t) = \sum_i W_{ji}^{\text{in}} x_i(t - d_{ji}^{\text{in}}) + \sum_i W_{ji}^{\text{rec}} z_i(t - d_{ji}^{\text{rec}}) \tag{4}$$

where $W_{ji}^{\text{in}}$ and $W_{ji}^{\text{rec}}$ denote respectively the input and the recurrent synaptic weights and $d_{ji}^{\text{in}}$ and $d_{ji}^{\text{rec}}$ the corresponding synaptic delays.

#### LIF neurons with SFA

The SFA is realized by replacing the fixed threshold $v_{\text{th}}$ with the adaptive threshold $A_j(t)$, which follows the dynamics (reproducing *Equation (1)* for arbitrary $\delta t$):

$$\begin{aligned} A_j(t) &= v_{\text{th}} + \beta a_j(t), \\ a_j(t + \delta t) &= \rho_j a_j(t) + (1 - \rho_j) z_j(t) \delta t. \end{aligned} \tag{5}$$

Now, the parameter $\rho_j$ is given by $\rho_j = \exp\left(\frac{-\delta t}{\tau_{a,j}}\right)$. In all our simulations, $\delta t$ was set to 1 ms.

The spiking output of LIF neuron with SFA $j$ is then defined by $z_j(t) = H\left(\frac{V_j(t) - A_j(t)}{A_j(t)}\right)\frac{1}{\delta t}$.

Adaptation time constants of neurons with SFA were chosen to match the task requirements while still conforming to the experimental data from rodents (*Allen Institute, 2018b*; *Pozzorini et al., 2013*; *Pozzorini et al., 2015*; *Mensi et al., 2012*). For an analysis of the impact of the adaptation time constants on the performance, see *Table 1*.

#### LIF neurons with activity-dependent increase in excitability: ELIF neurons

There exists experimental evidence that some neurons fire for the same stimulus more for a repetition of the same sensory stimulus. We refer to such neurons as ELIF neurons since they are becoming more excitable. Such repetition enhancement was discussed, for example, in *Tartaglia et al., 2014*. But to the best of our knowledge, it has remained open whether repetition enhancement is a

network effect, resulting, for example, from a transient depression of inhibitory synapses onto the cell that is caused by postsynaptic firing (*Kullmann et al., 2012*), or a result of an intrinsic firing property of some neurons. We used a simple model for ELIF neurons that is dual to the above-described LIF neuron model with SFA: the threshold is lowered through each spike of the neuron, and then decays exponentially back to its resting value. This can be achieved by using a negative value for β in *Equation (1)*.

## Models for short-term plasticity (STP) of synapses

We modeled the STP dynamics according to the classical model of STP in *Mongillo et al., 2008*. The STP dynamics in discrete time, derived from the equations in *Mongillo et al., 2008*, are as follows:

$$u'_{ji}(t+\delta t) = \exp\left(\frac{-\delta t}{F}\right) u'_{ji}(t) + U_{ji}(1 - u_{ji}(t))z_i(t)\delta t, \tag{6}$$

$$u_{ji}(t+\delta t) = U_{ji} + u'_{ji}(t), \tag{7}$$

$$r'_{ji}(t+\delta t) = \exp\left(\frac{-\delta t}{D}\right) r'_{ji}(t) + u_{ji}(t)(1 - r'_{ji}(t))z_i(t)\delta t, \tag{8}$$

$$r_{ji}(t+\delta t) = 1 - r'_{ji}(t), \tag{9}$$

$$W^{STP}_{ji}(t+\delta t) = W^{\mathrm{rec}}_{ji} u_{ji}(t) r_{ji}(t), \tag{10}$$

where $z_i(t)$ is the spike train of the presynaptic neuron and $W^{\mathrm{rec}}_{ji}$ scales the synaptic efficacy of synapses from neuron $i$ to neuron $j$. Networks with STP were constructed from LIF neurons with the weight $W^{\mathrm{rec}}_{ji}$ in *Equation (4)* replaced by the time-dependent weight $W^{STP}_{ji}(t)$.

STP time constants of facilitation-dominant and depression-dominant network models were based on the values of experimental recordings in *Wang et al., 2006* of PFC-E1 ($D = 194 \pm 18$ ms, $F = 507 \pm 37$ ms, $U = 0.28 \pm 0.02$) and PFC-E2 ($D = 671 \pm 17$ ms, $F = 17 \pm 5$ ms, $U = 0.25 \pm 0.02$) synapse types, respectively. Recordings in *Wang et al., 2006* were performed in the medial prefrontal cortex of young adult ferrets. In the sMNIST task for the depression-dominant network model (STP-D), we used values based on PFC-E2, and for facilitation-dominant network model (STP-F) we used values based on PFC-E1 (see sMNIST task section below). For the STORE-RECALL task, we trained the network with the data-based time constants based on PFC-E2 and PFC-E1 and also an extended time constants variant where both facilitation and depression time constants were equally scaled up until the larger time constant matched the requirement of the task (see One-dimensional STORE-RECALL task section below).

## Weight initialization

Initial input and recurrent weights were drawn from a Gaussian distribution $W_{ji} \sim \frac{w_0}{\sqrt{n_{\mathrm{in}}}} \mathcal{N}(0,1)$, where $n_{\mathrm{in}}$ is the number of afferent neurons and $\mathcal{N}(0,1)$ is the zero-mean unit-variance Gaussian distribution and $w_0 = \frac{1\,\mathrm{Volt}}{R_m}\delta t$ is a normalization constant (*Bellec et al., 2018a*). In the default setting, it is possible for neurons to have both positive and negative outgoing weights, also to change their sign during the optimization process. See Section 2 of Appendix 1 for more results with sparse connectivity and enforcement of Dale's law using deep rewiring (*Bellec et al., 2018b*).

## Sigmoid and softmax functions

In the STORE-RECALL task (1- and 20-dimensional), the sigmoid function was applied to the neurons in the output layer. The sigmoid function is given by

$$\sigma(x) = \frac{1}{1 + \mathrm{e}^{-x}}, \tag{11}$$

where $x$ represents a real-valued variable. The result, bounded to $[0, 1]$ range, is then thresholded at the value of 0.5 to obtain the final predictions – neuron active or not. More precisely, the neuron is active if $\sigma(x) \geq 0.5$, otherwise it is not.

The softmax function (used in tasks sMNIST, 12AX, Duplication/Reversal) is given by

$$\text{Softmax}(x_i) = \frac{e^{x_i}}{\sum_{j=1}^{m} e^{x_j}}, \tag{12}$$

where $x_i$ is a real-valued output of neuron $i$ in the output layer with $m$ neurons. The final prediction after applying the softmax is then obtained by taking the maximum of all values calculated for each neuron in the output layer.

## Training methods
### BPTT

In artificial recurrent neural networks, gradients can be computed with *BPTT* (*Mozer, 1989*; *Robinson and Fallside, 1987*; *Werbos, 1988*). In SNNs, complications arise from the non-differentiability of the output of spiking neurons. In our discrete-time simulation, this is formalized by the discontinuous step function $H$ arising in the definition of the spike variable $z_j(t)$. All other operations can be differentiated exactly with *BPTT*. For feedforward artificial neural networks using step functions, a solution was to use a pseudo-derivative $H'(x) := \max\{0, 1 - |x|\}$ (*Esser et al., 2016*), but this method is unstable with recurrently connected neurons. It was found in *Bellec et al., 2018a* that dampening this pseudo-derivative with a factor $\gamma < 1$ (typically $\gamma = 0.3$) solves that issue. Hence, we use the pseudo-derivative

$$\frac{dz_j(t)}{dv_j(t)} := \gamma \max\{0, 1 - |v_j(t)|\}, \tag{13}$$

where $v_j(t)$ denotes the normalized membrane potential $v_j(t) = \frac{V_j(t) - A_j(t)}{A_j(t)}$. Importantly, gradients can propagate in adaptive neurons through many time steps in the dynamic threshold without being affected by the dampening.

Unless stated otherwise, the input, the recurrent, and the readout layers were fully connected and the weights were trained simultaneously.

### e-prop

In the 12AX task, the networks were trained using the biologically plausible learning method *random e-prop* (*Bellec et al., 2020*) in addition to *BPTT*.

## Tasks
### One-dimensional STORE-RECALL task

The input to the network consisted of 40 input neurons: 10 for STORE, 10 for RECALL, and 20 for population coding of a binary feature. Whenever a subpopulation was active, it would exhibit a Poisson firing with a frequency of 50 Hz. For experiments reported in *Figure 2D*, each input sequence consisted of 20 steps (200 ms each) where the STORE or the RECALL populations were activated with probability 0.09 interchangeably, which resulted in delays between the STORE-RECALL pairs to be in the range [200, 3600] ms. For experiments reported in *Table 1*, the input sequences of experiments with the expected delay of 2, 4, 8, and 16 s were constructed as a sequence of 20, 40, 80, and 120 steps, respectively, with each step lasting for 200 ms. For the experiment with expected delay of 200 ms, the input sequence consisted of 12 steps of 50 ms.

Networks were trained for 400 iterations with a batch size of 64 in *Table 1* and 128 in *Figure 2B*. We used Adam optimizer with default parameters and initial learning rate of 0.01, which was decayed every 100 iterations by a factor of 0.3. To avoid unrealistically high firing rates, the loss function contained a regularization term (scaled with coefficient 0.001) that minimizes the squared difference of the average firing rate of individual neurons from a target firing rate of 10 Hz. In *Figure 1D, E*, the weights were chosen by hand and not trained. The test performance was computed as the batch average over 2048 random input sequences.

Networks consisted of 60 recurrently connected neurons in all experiments except in *Figure 1D, E*, where only two neurons were used without recurrent connections. The membrane time constant was $\tau_m = 20$ ms, the refractory period 3 ms. In *Figure 1D, E*, the two LIF neurons with SFA had $\beta = 3$ mV and $\tau_a = 1200$ ms. In *Figure 2D*, for LIF with SFA and ELIF networks, we used $\beta = 1$ mV and $\beta = -0.5$ mV, respectively, with $\tau_a = 2000$ ms. *Table 1* defines the adaptation time constants and expected delay of the experiments in that section. To provide a fair comparison between STP and SFA models in *Figure 2D*, we train two variants of the STP model: one with the original parameters from *Wang et al., 2006* and another where we scaled up both $F$ and $D$ until the larger one reached 2000 ms, the same time constant used in the SFA model. The scaled up synapse parameters of STP-D network were $F = 51 \pm 15$ ms, $D = 2000 \pm 51$ ms, and $U = 0.25$, and of STP-F network $F = 2000 \pm 146$ ms, $D = 765 \pm 71$ ms, and $U = 0.28$. The data-based synapse parameters are described in the STP synapse dynamics section above. The baseline threshold voltage was 10 mV for all models except ELIF for which it was 20 mV and the two neurons in *Figure 1D, E* for which it was 5 mV. The synaptic delay was 1 ms. The input to the sigmoidal readout neurons were the neuron traces that were calculated by passing all the network spikes through a low-pass filter with a time constant of 20 ms.

## 20-Dimensional STORE-RECALL task

The input to the network consisted of commands STORE and RECALL, and 20 bits, which were represented by subpopulations of spiking input neurons. STORE and RECALL commands were represented by four neurons each. The 20 bits were represented by population coding where each bit was assigned four input neurons (two for value 0, and two for value 2). When a subpopulation was active, it would exhibit a Poisson firing with a frequency of 400 Hz. Each input sequence consisted of 10 steps (200 ms each) where a different population encoded bit string was shown during every step. Only during the RECALL period the input populations, representing the 20 bits, were silent. At every step, the STORE or the RECALL populations were activated interchangeably with probability 0.2, which resulted in the distribution of delays between the STORE-RECALL pairs in the range [200, 1600] ms.

To measure the generalization capability of a trained network, we first generated a test set dictionary of 20 unique feature vectors (random bit strings of length 20) that had at least a Hamming distance of 5 bits among each other. For every training batch, a new dictionary of 40 random bit strings (of length 20) was generated, where each string had a Hamming distance of at least 5 bits from any of the bit string in the test set dictionary. This way we ensured that, during training, the network never encountered any bit string similar to one from the test set.

Networks were trained for 4000 iterations with a batch size of 256 and stopped if the error on the training batch was below 1%. We used Adam optimizer (*Kingma and Ba, 2014*) with default parameters and initial learning rate of 0.01, which is decayed every 200 iterations by a factor of 0.8. We also used learning rate ramping, which, for the first 200 iterations, monotonically increased the learning rate from 0.00001 to 0.01. The same firing rate regularization term was added to the loss as in the one-dimensional STORE-RECALL setup (see above). To improve convergence, we also included an entropy component to the loss (scaled with coefficient 0.3), which was computed as the mean of the entropies of the outputs of the sigmoid neurons. The test performance was computed as average over 512 random input sequences.

We trained SNNs with and without SFA, consisting of 500 recurrently connected neurons. The membrane time constant was $\tau_m = 20$ ms, and the refractory period was 3 ms. Adaptation parameters were $\beta = 4$ mV and $\tau_a = 800$ ms with baseline threshold voltage 10 mV. The synaptic delay was 1 ms. The same sigmoidal readout neuron setup was used as in the one-dimensional STORE-RECALL setup (see above).

We ran five training runs with different random seeds (initializations) for both SNNs with and without SFA. All runs of the SNN with SFA network converged after ~ 3600 iterations to a training error below 1%. At that point we measured the accuracy on 512 test sequences generated using the previously unseen test bit strings, which resulted in test accuracy of 99.09% with a standard deviation of 0.17%. The LIF network was not able to solve the task in any of the runs (all runs resulted in 0% training and test accuracy with zero standard deviation). On the level of individual feature recall accuracy, the best one out of five training runs of the LIF network was able to achieve 49% accuracy, which is

the chance level since individual features are binary bits. In contrast, all SNNs with SFA runs had individual feature-level accuracy of above 99.99%.

## Decoding memory from the network activity

We trained a support vector machine (SVM) to classify the stored memory content from the network spiking activity in the step before the RECALL (200 ms before the start of RECALL command). We performed a cross-validated grid-search to find the best hyperparameters for the SVM, which included kernel type {linear, polynomial, RBF} and penalty parameter $C$ of the error term {0.1, 1, 10, 100, 1000}. We trained SVMs on test batches of the five different training runs of 20-dimensional STORE-RECALL task. SVMs trained on the period preceding the RECALL command of a test batch achieved an average of 4.38% accuracy with a standard deviation of 1.29%. In contrast, SVMs trained on a period during the RECALL command achieved an accuracy of 100%. This demonstrates that the memory stored in the network is not decodable from the network firing activity before the RECALL input command.

Additionally, analogous to the experiments of *Wolff et al., 2017*, we trained SVMs on network activity during the encoding (STORE) period and evaluated them on the network activity during reactivation (RECALL), and vice versa. In both scenarios, the classifiers were not able to classify the memory content of the evaluation period (0.0% accuracy).

## sMNIST task

The input consisted of sequences of 784 pixel values created by unrolling the handwritten digits of the MNIST dataset, one pixel after the other in a scanline manner as indicated in *Appendix 1—figure 3A*. We used 1 ms presentation time for each pixel gray value. Each of the 80 input neurons was associated with a particular threshold for the gray value, and this input neuron fired whenever the gray value crossed its threshold in the transition from the previous to the current pixel.

Networks were trained for 36,000 iterations using the Adam optimizer with batch size 256. The initial learning rate was 0.01, and every 2500 iterations the learning rate was decayed by a factor of 0.8. The same firing rate regularization term was added to the loss as in the STORE-RECALL setup (see above) but with the scaling coefficient of 0.1.

All networks consisted of 220 neurons. Network models labeled LIF with SFA and ELIF in the *Figure 2C* had 100 neurons out of 220 with SFA or transient excitability, respectively. The network with SFA had 100 neurons out of 220 with SFA and the rest without. The neurons had a membrane time constant of $\tau_m = 20$ ms, a baseline threshold of $v_{\text{th}} = 10$ mV, and a refractory period of 5 ms. LIF neurons with SFA and ELIF neurons had the adaptation time constant $\tau_a = 700$ ms with adaptation strength $\beta = 1.8$ mV and –0.9 mV, respectively. The synaptic delay was 1 ms. Synapse parameters were $F = 20$ ms, $D = 700$ ms, and $U = 0.2$ for the STP-D model, and $F = 500$ ms, $D = 200$ ms, and $U = 0.2$ for the STP-F model. The output of the SNN was produced by the softmax of 10 linear output neurons that received the low-pass filtered version of the spikes from all neurons in the network, as shown in the bottom row of *Appendix 1—figure 3B*. The low-pass filter had a time constant of 20 ms. For training the network to classify into one of the 10 classes, we used cross-entropy loss computed between the labels and the softmax of output neurons.

## The 12AX task

The input for each training and testing episode consisted of a sequence of 90 symbols from the set {1,2,A,B,C,X,Y,Z}. A single episode could contain multiple occurrences of digits 1 or 2 (up to 23), each time changing the target sequence (A...X or B...Y) after which the network was supposed to output R. Each digit could be followed by up to 26 letters before the next digit appeared. More precisely, the following regular expression describes the string that was produced: [12][ABCXYZ]{1,10} ((A[CZ]{0,6}X|B[CZ]{0,6}Y)|([ABC][XYZ])){1,2}. Each choice in this regular expression was made randomly.

The network received spike trains from the input population of spiking neurons, producing Poisson spike trains. Possible input symbols were encoded using 'one-hot encoding' scheme. Each input symbol was signaled through a high firing rate of a separate subset of five input neurons for 500 ms. The output consisted of two readout neurons, one for L, one for the R response. During each 500 ms time window, the input to these readouts was the average activity of neurons in the SNN

during that time window. The final output symbol was based on which of the two readouts had the maximum value.

The neurons had a membrane time constant of $\tau_m = 20$ ms, a baseline threshold $v_{\text{th}} = 30$ mV, a refractory period of 5 ms, and synaptic delays of 1 ms. LIF neurons with SFA had an adaptation strength of $\beta = 1.7$ mV, and adaptation time constants were chosen uniformly from $[1, 13500]$ ms.

A cross-entropy loss function was used to minimize the error between the softmax applied to the output layer and targets, along with a regularization term (scaled with coefficient 15) that minimized the squared difference of average firing rate between individual neurons and a target firing rate of 10 Hz. The SNN was trained using the Adam optimizer for 10,000 iterations with a batch size of 20 episodes and a fixed learning rate of 0.001. An episode consisted of 90 steps, with between 4 and 23 tasks generated according to the task generation procedure described previously. We trained the network consisting of 200 LIF neurons (100 with and 100 without SFA) with *BPTT* using five different network initializations, which resulted in an average test success rate of 97.79 with a standard deviation of 0.42%.

In the experiments where the fraction of neurons with SFA varied, the network with 200 LIF neurons with SFA (i.e., all LIF neurons with SFA) achieved a success rate of 72.01% with a standard deviation of 36.15%, whereas the network with only 20 LIF neurons with SFA and 180 LIF neurons without SFA achieved a success rate of 95.39% with a standard deviation of 1.55%. The network consisting of 200 LIF neurons without SFA (i.e., all neurons without SFA) was not able to solve the task, and it achieved a success rate of 0.39% with a standard deviation of 0.037%. Each success rate reported is an average calculated over five different network initializations.

The network consisting of 100 LIF neurons with and 100 LIF neurons without SFA, trained with *random e-prop*, resulted in an average test success rate of 92.89% with a standard deviation of 0.75% (average over five different network initializations).

## Symbolic computation on strings of symbols (Duplication/Reversal task)

The input to the network consisted of 35 symbols: 31 symbols represented symbols from the English alphabet {a, b, c, d, ... x, y, z, A, B, C, D, E}, one symbol was for 'end-of-string' (EOS) '*', one for cue for the output prompt '?', and two symbols to denote whether the task command was duplication or reversal. Each of the altogether 35 input symbols were given to the network in the form of higher firing activity of a dedicated population of 5 input neurons outside of the SNN ('one-hot encoding'). This population of input neurons fired at a 'high' rate (200 Hz) to encode 1, and at a 'low' rate (2 Hz) otherwise. The network output was produced by linear readouts (one per potential output symbol, each with a low-pass filter with a time constant of 250 ms) that received spikes from neurons in the SNN (see the row 'Output' in *Figure 4A*). The final output symbol was selected using the readout that had the maximum value at the end of each 500 ms time window (a softmax instead of the hard argmax was used during training), mimicking WTA computations in neural circuits of the brain (*Chettih and Harvey, 2019*) in a qualitative manner.

The network was trained to minimize the cross-entropy error between the softmax applied to the output layer and targets. The loss function contained a regularization term (scaled with coefficient 5) that minimizes the squared difference of average firing rate between individual neurons and a target firing rate of 20 Hz.

The training was performed for 50,000 iterations, with a batch size of 50 episodes. We used Adam optimizer with default parameters and a fixed learning rate of 0.001. Each symbol was presented to the network for a duration of 500 ms. The primary metric we used for measuring the performance of the network was success rate, which was defined as the percentage of episodes where the network produced the full correct output for a given string, that is, all the output symbols in the episode had to be correct. The network was tested on 50,000 previously unseen strings.

The network consisted of 192 LIF neurons with SFA and 128 LIF neurons without SFA. All the neurons had a membrane time constant of $\tau_m = 20$ ms, a baseline threshold $v_{\text{th}} = 30$ mV, a refractory period of 5 ms, and a synaptic delay of 1 ms. LIF neurons with SFA in the network had an adaptation strength of $\beta = 1.7$ mV. It was not necessary to assign particular values to adaptation time constants of firing thresholds of neurons with SFA; we simply chose them uniformly randomly to be between 1 ms and 6000 ms, mimicking the diversity of SFA effects found in the neocortex (*Allen Institute, 2018b*) in a qualitative manner. All other parameters were the same as in the other experiments. We

trained the network using five different network initializations (seeds) and tested it on previously unseen strings. Average test success rate was 95.88% with standard deviation 1.39%.

## Analysis of spiking data for Duplication/Reversal task

We used three-way ANOVA to analyze if a neuron's firing rate is significantly affected by *task*, *serial position* in the sequence, *symbol identity*, or combination of these (similar to *Lindsay et al., 2017*). In such a multifactorial experiment, factors are crossed with each other, and we refer to these factors as 'conditions.' For two possible tasks, 5 possible positions in the input sequence, and 31 possible symbols, there are $2*5*31 = 310$ different conditions. The analysis was performed on the activity of the neurons of the trained SNN during 50,000 test episodes. From each episode, a serial position from the input period was chosen randomly, and hence each episode could be used only once, that is, as one data point. This was to make sure that each entry in the three-way ANOVA was completely independent of other entries since the neuron's activity within an episode is highly correlated. Each data point was labeled with the corresponding triple of (task type, serial position, symbol identity). To ensure that the dataset was balanced, the same number of data points per particular combination of conditions was used, discarding all the excess data points, resulting in a total of 41,850 data points – 135 data points per condition, that is, 135 repeated measurements for each condition and per neuron, but with no carryover effects for repetitions per neuron since the internal state variables of a neuron are reset between episodes. In such a scenario, neurons can be seen as technical replicates. For the analysis, neurons whose average firing rate over all episodes (for the input period) was lower than 2 Hz or greater than 60 Hz were discarded from the analysis to remove large outliers. This left 279 out of the 320 neurons. To categorize a neuron as selective to one or more conditions, or combination of conditions, we observed p-values obtained from three-way ANOVA and calculated the effect size $\omega^2$ for each combination of conditions. If the p-value was less than 0.001 and $\omega^2$ greater than 0.14 for a particular combination of conditions, the neuron was categorized as selective to that combination of conditions. The $\omega^2$ threshold of 0.14 was suggested by *Field, 2013* to select large effect sizes. Each neuron can have a large effect size for more than one combination of conditions. Thus, the values shown in *Figure 4D* sum to a value greater than 1. The neuron shown in *Figure 4E* had the most prominent selectivity for the combination of Task $\times$ Position $\times$ Symbol, with $\omega^2 = 0.394$ and $p<0.001$. The neuron shown in *Figure 4F* was categorized as selective to a combination of Position $\times$ Symbol category, with $\omega^2 = 0.467$ and $p<0.001$. While the three-way ANOVA tells us if a neuron is selective to a particular combination of conditions, it does not give us the exact task/symbol/position that the neuron is selective to. To find the specific task/symbol/position that the neuron was selective to, Welch's t-test was performed, and a particular combination with maximum t-statistic and $p<0.001$ was chosen to be shown in *Figure 4E, F*.

## Acknowledgements

We would like to thank Pieter Roelfsema and Christopher Summerfield for detailed comments on an earlier version of the manuscript. This research was partially supported by the Human Brain Project (Grant Agreement number 785907 and 945539), the SYNCH project (Grant Agreement number 824162) of the European Union, and under partial support by the Austrian Science Fund (FWF) within the ERA-NET CHIST-ERA programme (project SMALL, project number I 4670-N). We gratefully acknowledge the support of NVIDIA Corporation with the donation of the Quadro P6000 GPU used for this research. Computations were carried out on the Human Brain Project PCP Pilot Systems at the Juelich Supercomputing Centre, which received co-funding from the European Union (Grant Agreement number 604102) and on the Vienna Scientific Cluster (VSC).

## Additional information

### Funding

| Funder | Grant reference number | Author |
| --- | --- | --- |
| Horizon 2020 Framework Programme | Human Brain Project 785907 | Darjan Salaj<br>Anand Subramoney<br>Guillaume Bellec |

| Horizon 2020 Framework Programme | Human Brain Project 945539 | Darjan Salaj<br>Anand Subramoney<br>Guillaume Bellec<br>Wolfgang Maass |
|---|---|---|
| Horizon 2020 Framework Programme | SYNCH project 824162 | Ceca Kraisnikovic<br>Robert Legenstein |
| FWF Austrian Science Fund | ERA-NET CHIST-ERA programme (project SMALL project number I 4670-N) | Robert Legenstein |

The funders had no role in study design, data collection and interpretation, or the decision to submit the work for publication.

## Author contributions

Darjan Salaj, Conceptualization, Data curation, Software, Formal analysis, Validation, Investigation, Visualization, Methodology, Writing - original draft, Writing - review and editing; Anand Subramoney, Conceptualization, Data curation, Software, Formal analysis, Supervision, Validation, Investigation, Methodology, Writing - original draft, Writing - review and editing; Ceca Kraisnikovic, Data curation, Software, Formal analysis, Validation, Investigation, Visualization, Writing - original draft, Writing - review and editing; Guillaume Bellec, Conceptualization, Software, Supervision, Methodology, Writing - review and editing; Robert Legenstein, Resources, Supervision, Funding acquisition, Methodology, Writing - original draft, Project administration, Writing - review and editing; Wolfgang Maass, Conceptualization, Resources, Supervision, Funding acquisition, Investigation, Methodology, Writing - original draft, Writing - review and editing

## Author ORCIDs

Darjan Salaj https://orcid.org/0000-0001-9183-5852
Anand Subramoney https://orcid.org/0000-0002-7333-9860
Ceca Kraisnikovic https://orcid.org/0000-0003-0906-920X
Guillaume Bellec https://orcid.org/0000-0001-7568-4994
Robert Legenstein https://orcid.org/0000-0002-8724-5507
Wolfgang Maass https://orcid.org/0000-0002-1178-087X

## Decision letter and Author response

Decision letter https://doi.org/10.7554/eLife.65459.sa1
Author response https://doi.org/10.7554/eLife.65459.sa2

# Additional files

## Supplementary files

• Transparent reporting form

## Data availability

An implementation of the network model in Tensorflow/Python is available at https://github.com/IGITUGraz/LSNN-official (copy archived at https://archive.softwareheritage.org/swh:1:rev:a9158a3540da92ae51c46a3b7abd4eae75a2bb86). The sMNIST dataset is available at https://www.tensorflow.org/datasets/catalog/mnist. The google speech commands dataset is available at https://storage.cloud.google.com/download.tensorflow.org/data/speech_commands_v0.02.tar.gz. The code for experiments presented in the main paper are available at https://github.com/IGITUGraz/Spike-Frequency-Adaptation-Supports-Network-Computations.

The following datasets were generated:

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

## Appendix 1

### Autocorrelation-based intrinsic time scale of neurons trained on STORE-RECALL task

We wondered whether the adaptive firing threshold of LIF neurons with SFA affects the autocorrelation function of their firing activity – termed intrinsic time scale in *Wasmuht et al., 2018*. We tested this for an SNN consisting of 200 LIF neurons without and 200 LIF neurons with SFA that was trained to solve a one-dimensional version of the STORE-RECALL task. It turned out that during the delay between STORE and RECALL these intrinsic time constants were in the same range as those measured in the monkey cortex (see Figure 1C in *Wasmuht et al., 2018*). Furthermore, neurons of the trained SNN exhibited very similar distributions of these time constants (see *Appendix 1—figure 1*), suggesting that these intrinsic time constants are determined largely by their network inputs, and less by the neuron type.

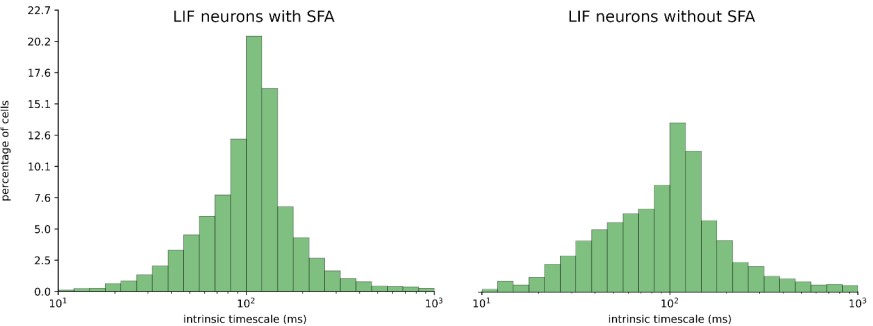

**Appendix 1—figure 1.** Histogram of the intrinsic time scale of neurons trained on STORE-RECALL task. We trained 64 randomly initialized spiking neural networks (SNNs) consisting of 200 leaky integrate-and-fire (LIF) neurons with and 200 without spike frequency adaptation (SFA) on the single-feature STORE-RECALL task. Measurements of the intrinsic time scale were performed according to *Wasmuht et al., 2018* on the spiking data of SNNs solving the task after training. Averaged data of all 64 runs is presented in the histogram. The distribution is very similar for neurons with and without SFA.

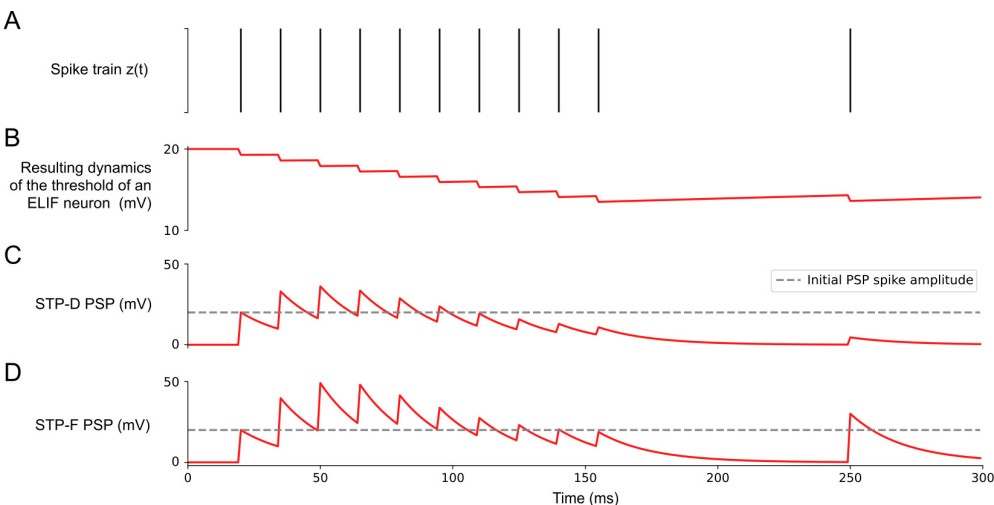

**Appendix 1—figure 2.** Illustration of models for an inversely adapting enhanced-excitability LIF (ELIF) neuron, and for short-term synaptic plasticity. (**A**) Sample spike train. (**B**) The resulting evolution of firing threshold for an inversely adapting neuron (ELIF neuron). (**C, D**) The resulting

*Appendix 1—figure 2 continued on next page*

*Appendix 1—figure 2 continued*

evolution of the amplitude of postsynaptic potentials (PSPs) for spikes of the presynaptic neuron for the case of a depression-dominant (STP-D: D >> F) and a facilitation-dominant (STP-F: F >> D) short-term synaptic plasticity.

## sMNIST task with sparsely connected SNN obeying Dale's law

This task has originally been used as a temporal processing benchmark for ANNs and has successfully been solved with the LSTM type of ANNs (*Hochreiter and Schmidhuber, 1997*). LSTM units store information in registers – like a digital computer – so that the stored information cannot be perturbed by ongoing network activity. Networks of LSTM units or variations of such units have been widely successful in temporal processing and reach the level of human performance for many temporal computing tasks.

Since LSTM networks also work well for tasks on larger time scales, for comparing SNNs with LSTM networks, we used a version of the task with 2 ms presentation time per pixel, thereby doubling the length of sequences to be classified to 1568 ms. Gray values of pixels were presented to the LSTM network simply as analog values. A trial of a trained SNN with SFA (with an input sequence that encodes a handwritten digit '3' using population rate coding) is shown in *Appendix 1—figure 3B*. The top row of *Figure 3B* shows a version where the gray value of the currently presented pixel is encoded by population coding, through the firing probability of 80 input neurons. Somewhat better performance was achieved when each of the 80 input neurons was associated with a particular threshold for the gray value, and this input neuron fired whenever the gray value crossed its threshold in the transition from the previous to the current pixel (this input convention was used to produce the results below).

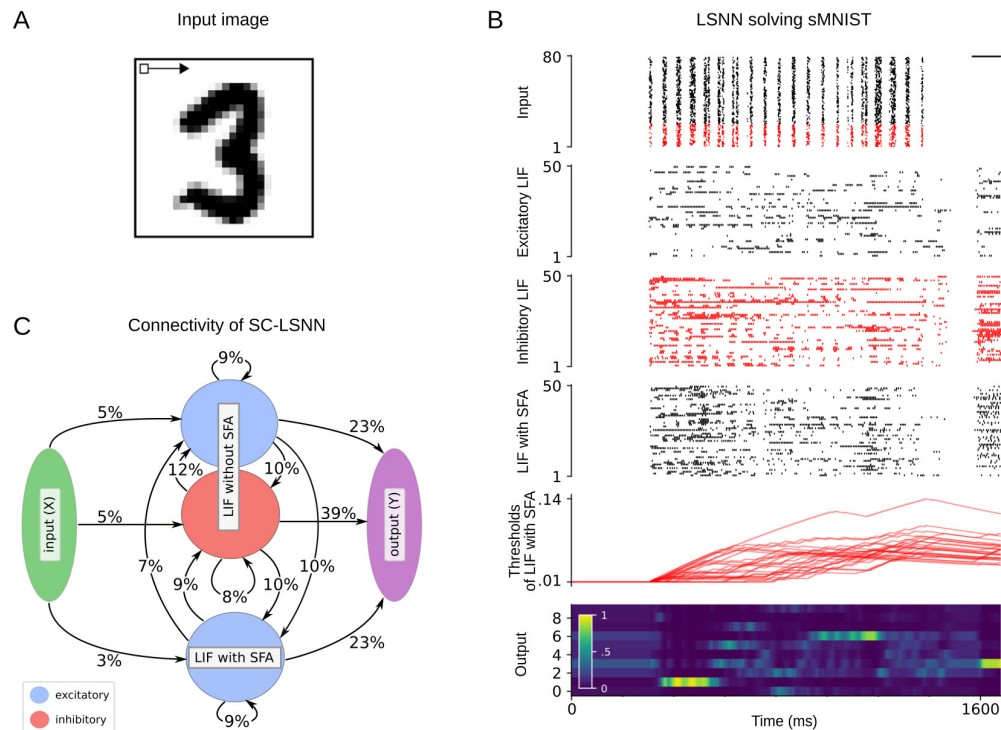

**Appendix 1—figure 3.** sMNIST time-series classification benchmark task. (**A**) Illustration of the pixel-wise input presentation of handwritten digits for sMNIST. (**B**) Rows top to bottom: input encoding for an instance of the sMNIST task, network activity, and temporal evolution of firing thresholds for randomly chosen subsets of neurons in the SC-SNN, where 25% of the leaky integrate-and-fire (LIF) neurons were inhibitory (their spikes are marked in red). The light color of the readout neuron for digit '3' around 1600 ms indicates that this input was correctly classified. (**C**) Resulting connectivity

*Appendix 1—figure 3 continued on next page*

graph between neuron populations of an SC-SNN after backpropagation through time (BPTT) optimization with DEEP R on sMNIST task with 12% global connectivity limit.

Besides a fully connected network of LIF neurons with SFA, we also tested the performance of a variant of the model, called SC-SNN, that integrates additional constraints of SNNs in the brain: it is sparsely connected (12% of possible connections are present) and consists of 75% excitatory and 25% inhibitory neurons that adhere to Dale's law. By adapting the sparse connections with the rewiring method in *Bellec et al., 2018a* during *BPTT* training, the SC-SNN was able to perform even better than the fully connected SNN of LIF neurons with SFA. The resulting architecture of the SC-SNN is shown in *Appendix 1—figure 3C*. Its activity of excitatory and inhibitory neurons, as well as the time courses of adaptive thresholds for (excitatory) LIF neurons with SFA of the SC-SNN, is shown in *Appendix 1—figure 3B*. In this setup, the SFA had $\tau_a = 1400$ ms. When we used an SNN with SFA, we improved the accuracy on this task to 96.4%, which approaches the accuracy of the artificial LSTM model that reached the accuracy of 98.0%.

We also trained a liquid state machine version of the SNN model with SFA where only the readout neurons are trained. This version of the network reached the accuracy of $63.24 \pm 1.48\%$ over five independent training runs.

## Google Speech Commands

We trained SNNs with and without SFA on the keyword-spotting task with Google Speech Commands Dataset (*Warden, 2018*) (v0.02). The dataset consists of 105,000 audio recordings of people saying 30 different words. Fully connected networks were trained to classify audio recordings, which were clipped to 1 s length, into one of 12 classes (10 keywords, as well as 2 special classes for silence and unknown words; the remaining 20 words had to be classified as 'unknown'). Comparison of the maximum performance of trained spiking networks against state-of-the-art artificial recurrent networks is shown in *Table 1*. Averaging over five runs, the SNN with SFA reached $90.88 \pm 0.22\%$, and the SNN without SFA reached $88.79 \pm 0.16\%$ accuracy. Thus an SNN without SFA can already solve this task quite well, but the inclusion of SFA halves the performance gap to the published state of the art in machine learning. The only other report on a solution to this task with spiking networks is *Zenke and Vogels, 2020*. There the authors train a network of LIF neurons using surrogate gradients with *BPTT* and achieve $85.3 \pm 0.3\%$ accuracy on the full 35 classes setup of the task. In this setup, the SNN with SFA reached $88.5 \pm 0.16\%$ test accuracy.

**Appendix 1—table 1.** Google Speech Commands.
Accuracy of the spiking network models on the test set compared to the state-of-the-art artificial recurrent model reported in *Kusupati et al., 2018*. Accuracy of the best out of five simulations for spiking neural networks (SNNs) is reported. SFA: spike frequency adaptation.

| Model | Test accuracy (%) |
| --- | --- |
| FastGRNN-LSQ (*Kusupati et al., 2018*) | 93.18 |
| SNN with SFA | 91.21 |
| SNN | 89.04 |

Features were extracted from the raw audio using the Mel Frequency Cepstral Coefficient (MFCC) method with 30 ms window size, 1 ms stride, and 40 output features. The network models were trained to classify the input features into one of the 10 keywords (yes, no, up, down, left, right, on, off, stop, go) or to two special classes for silence or unknown word (where the remainder of 20 recorded keywords are grouped). The training, validation, and test set were assigned 80, 10, and 10% of data respectively while making sure that audio clips from the same person stay in the same set.

All networks were trained for 18,000 iterations using the Adam optimizer with batch size 100. The output spikes of the networks were averaged over time, and the linear readout layer was applied to those values. During the first 15,000 iterations, we used a learning rate of 0.001 and for the last

3000, we used a learning rate of 0.0001. The loss function contained a regularization term (scaled with coefficient 0.001) that minimizes the squared difference of average firing rate between individual neurons and a target firing rate of 10 Hz.

Both SNNs with and without SFA consisted of 2048 fully connected neurons in a single recurrent layer. The neurons had a membrane time constant of $\tau_m = 20$ ms, the adaptation time constant of SFA was $\tau_a = 100$ ms, and adaptation strength was $\beta = 2$ mV. The baseline threshold was $v_{\text{th}} = 10$ mV, and the refractory period was 2 ms. The synaptic delay was 1 ms.

## Delayed-memory XOR

We also tested the performance of SNNs with SFA on a previously considered benchmark task, where two items in the working memory have to be combined nonlinearly: the delayed-memory XOR task (*Huh and Sejnowski, 2018*). The network is required to compute the exclusive-or operation on the history of input pulses when prompted by a go-cue signal (see *Appendix 1—figure 4*).

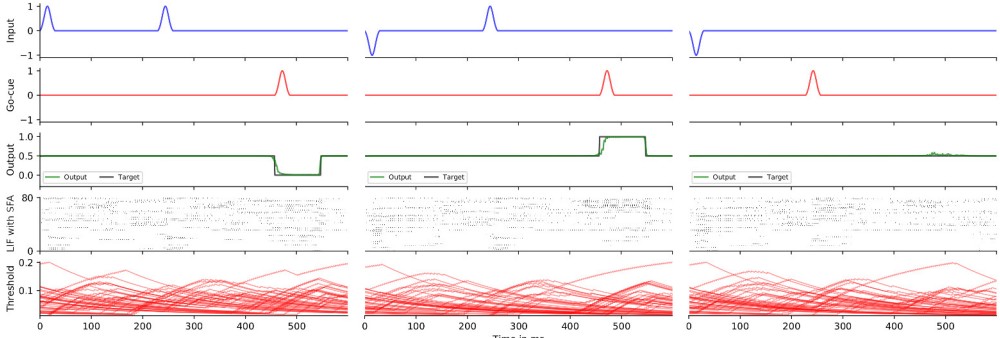

**Appendix 1—figure 4.** Delayed-memory XOR task. Rows top to bottom: input signal, go-cue signal, network readout, network activity, and temporal evolution of firing thresholds.

The network received on one input channel two types of pulses (up or down) and a go-cue on another channel. If the network received two input pulses since the last go-cue signal, it should generate the output '1' during the next go-cue if the input pulses were different or '0' if the input pulses were the same. Otherwise, if the network only received one input pulse since the last go-cue signal, it should generate a null output (no output pulse). Variable time delays are introduced between the input and go-cue pulses. The time scale of the task was 600 ms, which limited the delay between input pulses to 200 ms.

This task was solved in *Huh and Sejnowski, 2018*, without providing performance statistics, by using a type of neuron that has not been documented in biology – a non-leaky quadratic integrate and fire neuron. We are not aware of previous solutions by networks of LIF neurons. To compare and investigate the impact of SFA on network performance in the delayed-memory XOR task, we trained SNNs, with and without SFA, of the same size as in *Huh and Sejnowski, 2018* – 80 neurons. Across 10 runs, SNNs with SFA solved the task with $95.19 \pm 0.014\%$ accuracy, whereas the SNNs without SFA converged at lower $61.30 \pm 0.029\%$ accuracy.

The pulses on the two input channels were generated with 30 ms duration and the shape of a normal probability density function normalized in the range $[0, 1]$. The pulses were added or subtracted from the baseline zero input current at appropriate delays. The go-cue was always a positive current pulse. The six possible configurations of the input pulses (+, –, ++, —, +-, –+) were sampled with equal probability during training and testing.

Networks were trained for 2000 iterations using the Adam optimizer with batch size 256. The initial learning rate was 0.01, and every 200 iterations the learning rate was decayed by a factor of 0.8. The loss function contained a regularization term (scaled with coefficient 50) that minimizes the squared difference of the average firing rate of individual neurons from a target firing rate of 10 Hz. This regularization resulted in networks with a mean firing rate of 10 Hz where firing rates of individual neurons were spread in the range [1, 16] Hz.

Both SNNs with and without SFA consisted of 80 fully connected neurons in a single recurrent layer. The neurons had a membrane time constant of $\tau_m = 20$ ms, a baseline threshold $v_{\text{th}} = 10$ mV,

and a refractory period of 3 ms. SFA had an adaptation time constant of $\tau_a = 500$ ms and an adaptation strength of $\beta = 1$ mV. The synaptic delay was 1 ms. For training the network to classify the input into one of the three classes, we used the cross-entropy loss between the labels and the softmax of three linear readout neurons. The input to the linear readout neurons were the neuron traces that were calculated by passing all the network spikes through a low-pass filter with a time constant of 20 ms.

## 12AX task in a noisy network

As a control experiment, aimed at testing the robustness of the solution (performance as a function of the strength of added noise), we simulated the injection of an additional noise current into all LIF neurons (with and without SFA). The previously trained network (trained without noise) was reused and tested on a test set of 2000 episodes. In each discrete time step, the noise was added to the input current $I_j(t)$ (see **Equation (4)** in the main text), hence affecting the voltage of the neuron:

$$I_j(t) = \sum_i W_{ji}^{\text{in}} x_i(t - d_{ji}^{\text{in}}) + \sum_i W_{ji}^{\text{rec}} z_i(t - d_{ji}^{\text{rec}}) + I_{\text{noise}}, \qquad \text{(AE1)}$$

where $I_{\text{noise}}$ was drawn from a normal distribution with mean zero, and standard deviation $\sigma \in \{0.05, 0.075, 0.1, 0.2, 0.5\}$.

Performance of the network without noise was 97.85% (performance of one initialization of the network with 100 LIF neurons with SFA and 100 LIF neurons without SFA). During testing, including the noise current of mean zero and standard deviation $\sigma \in \{0.05, 0.075, 0.1, 0.2, 0.5\}$ led to the performance of 92.65, 89.05, 80.25, 27.25, and 0.25%, respectively. The network performance degrades gracefully up to a current of standard deviation of about 0.1.

For an illustration of the effect of noise, see **Appendix 1—figures 5** and **6**. There, we compare the output spikes, adaptive threshold, and membrane voltage of one neuron with noise current to the versions without noise. The shown simulations started from exactly the same initial condition and noise with standard deviation 0.05 (0.075) was injected only into the shown neuron (other neurons did not receive any noise current). One sees that even this weak noise current produces a substantial perturbation of the voltage, adaptive threshold, and spiking output of the neuron.

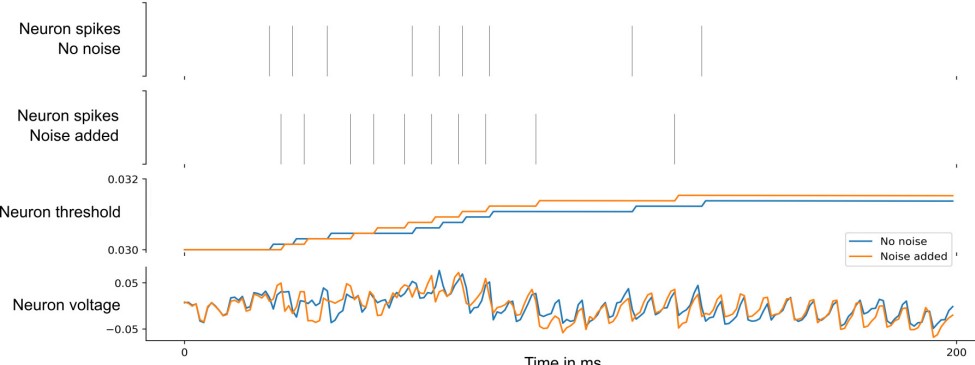

**Appendix 1—figure 5.** Effect of a noise current with zero mean and standard deviation 0.05 added to a single neuron in the 12AX task. Spike train of a single neuron without noise, followed by spike train in the presence of the noise, adaptive threshold of the neuron that corresponds to the spike train with no noise (shown in blue), spike train with noise present (shown in orange), and corresponding neuron voltages over the time course of 200 ms.

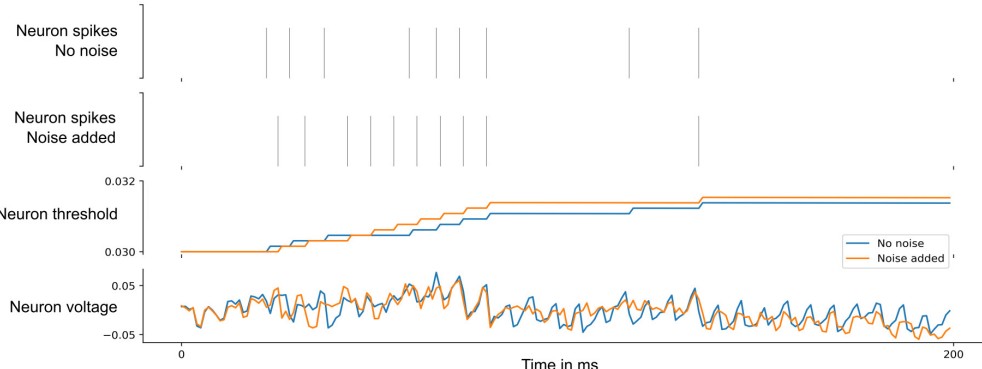

**Appendix 1—figure 6.** Effect of a noise current with zero mean and standard deviation 0.075 added to a single neuron in the network for the 12AX task. Spike train of a single neuron without noise, followed by spike train in the presence of the noise, adaptive threshold of the neuron that corresponds to the spike train with no noise (shown in blue), spike train with noise present (shown in orange), and corresponding neuron voltages over the time course of 200 ms.

## Duplication/Reversal task

A zoom-in for the rasters shown in *Figure 4A* (from the main text) is shown in *Appendix 1—figure 7* for the time period $3 - 4$ s.

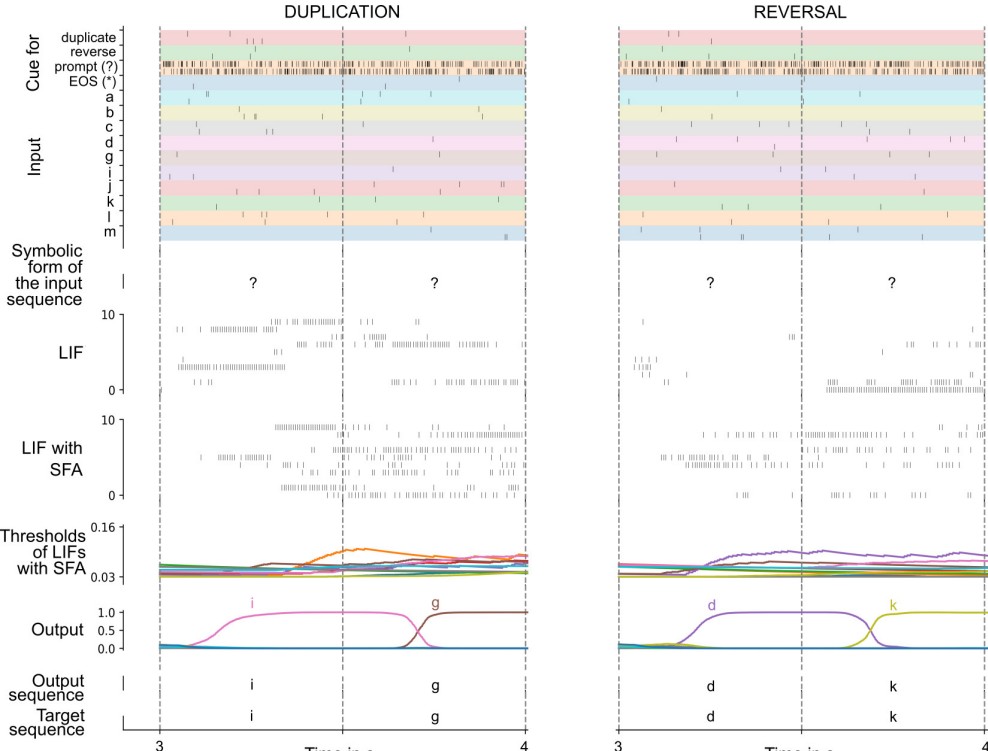

**Appendix 1—figure 7.** A zoom-in of the spike raster for a trial solving Duplication task (left) and Reversal task (right). A sample episode where the network carried out sequence duplication (left) and sequence reversal (right), shown for the time period of 3–4 ms (two steps after the start of network output). Top to bottom: spike inputs to the network (subset), sequence of symbols they encode, spike activity of 10 sample leaky integrate-and-fire (LIF) neurons (without and with spike frequency adaptation [SFA]) in the spiking neural network (SNN), firing threshold dynamics for these

*Appendix 1—figure 7 continued on next page*

*Appendix 1—figure 7 continued*

10 LIF neurons with SFA, activation of linear readout neurons, output sequence produced by applying argmax to them, and target output sequence.

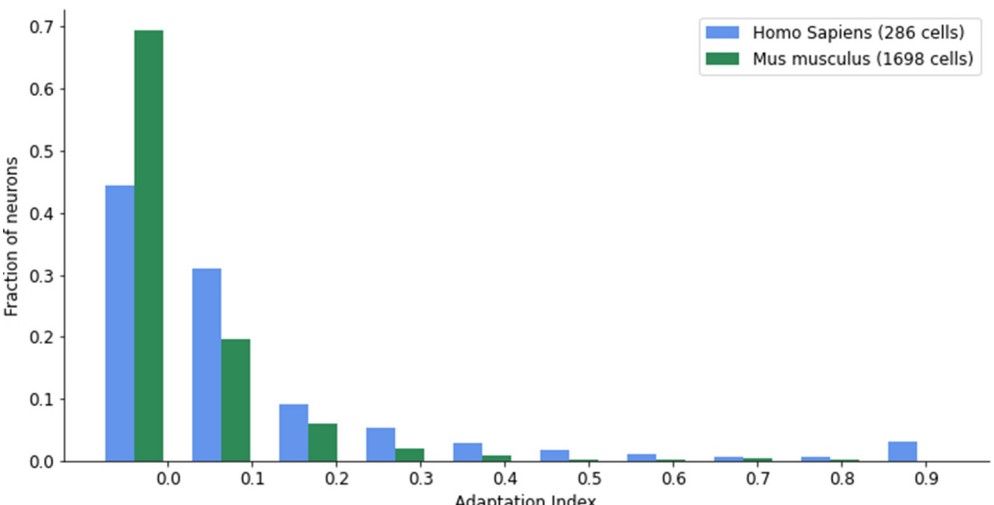

**Appendix 1—figure 8.** Distribution of adaptation index from Allen Institute cell measurements (*Allen Institute, 2018b*).

