## [Decision Letter]

**Acceptance summary:**

Although it is clear that the brain computes using spikes there are inherent obstacles to implementing many simple computational tasks using spiking networks, in part due to the short timescale associated with spiking events and synaptic potentials which makes computations involving short-term memory problematic. This paper demonstrates in simulations that longer timescale adaptation – spike frequency adaptation – can facilitate computations that in spiking networks by endowing them with a cellular form of short term memory. Interestingly, computational performance appears to benefit from heterogeneity in a network, with a subset of cells exhibiting adaptation. This demonstrates the importance of commonly overlooked physiological properties in implementing nontrivial computational tasks.

**Decision letter after peer review:**

Thank you for submitting your article "Spike frequency adaptation supports network computations on temporally dispersed information" for consideration by *eLife*. Your article has been reviewed by 2 peer reviewers, and the evaluation has been overseen by a Reviewing Editor and Timothy Behrens as the Senior Editor. The following individual involved in review of your submission has agreed to reveal their identity: Gabrielle Gutierrez (Reviewer #1).

Essential Revisions:

1) The authors need to rewrite the results and introduction with a broad biological audience in mind. As it stands, it targets readers who work on artificial neural networks and the results are essentially benchmarks against artificial neural networks. Many of the assumptions and connections to biology are not discussed adequately. In particular, key omissions and simplifications need to be discussed: the artificial nature of training, the lack of structure in the recurrent network, the likely differences between the statistics of spiking in the model vs what might be observed in an intact brain.

2) It would be helpful to lead the results with a simplified model that illustrates the negative imprinting mechanism directly. The results as they stand use rather complex models and provide largely observational evidence for the main claims of the paper. Since performance gains over other artificial neural networks are variable according to the task, some handle on when SFA is beneficial would help.

3) The relevance of SFA (as opposed to other slow timescale mechanisms) deserves fuller attention. The SFA characteristics are extracted from intracellular slice recordings where there is no substantial background activity. In an intact animal (with realistic spiking statistics) it is questionable whether the results will hold in a network where ongoing activity could maintain the SFA recovery variable close to steady-state. If this is the case, the conclusions need to be adequately tempered.

*Reviewer #1 (Recommendations for the authors):*The authors demonstrate the utility of spike frequency adaptation (SFA) in an artificial spiking neural network (SNN) for accomplishing tasks with a temporal dimension. They first show that a SNN with SFA can perform a simple store/recall task. They then test their model against tasks with increasing complexity. The authors find that in all of these tests, the SNN with SFA is able to match or outperform the most commonly implemented artificial neural network mechanisms which are also less biologically relevant. This study brings an awareness of the SFA mechanism to neural network modelers and it quantifies the improvement it brings to the performance of complex tasks. Furthermore, this study offers neurobiologists a new perspective on the SFA mechanism and its function in neural processing.

The most exciting strength of this paper is its multidisciplinary nature. It has the potential to drive innovation in the field of artificial neural networks (ANNs) by exploring how a known biological neural mechanism – SFA – can impact the performance of ANNs for solving temporally-complex tasks.

The multidisciplinary nature of this paper is a double-edged sword, however. At times there was a bit of a disconnect between the conclusions and the results. In most cases, the performance of the model against a canonical benchmark is rigorously quantified in a way that is standard in many neuroengineering papers. However, there were also multiple attempts to connect the model's performance to actual findings in the brain. I think that this is a worthwhile thing to do because it targets a broader readership that makes the study more relevant to the *eLife* audience; however, those comparisons were less rigorous than the ANN benchmark comparisons. This was particularly the case for the Negative Imprinting phenomenon and the emergent mixed-selectivity codes.

With regards to the rigorous benchmark comparisons presented, this paper could be made more accessible to the general neuroscience audience by putting the results and benchmark comparisons into a fuller context for those who are not as familiar with how well recent models have performed on these tasks. At times, the results showed very impressive improvement of the model over standard benchmarks while for other tasks the improvements appeared incremental. It would be helpful to have the performance improvements from the SFA model placed in the context of recent improvements in the canonical or state-of-the-art ANNs. This would allow the reader to know whether a small increase in performance represents an incremental improvement or a giant leap forward in the context of the history of ANNs.

Overall, there are a few main things I suggest the authors focus on revising. The figures and their captions could be made clearer in some places, as well as places in the main text that refer to specific parts of the figures. The other areas I think the authors should focus on are the last section of the Results, the Negative Imprinting Principle section, and the Discussion section.

The last section of the Results should probably be rewritten – especially the first paragraph. In general, in this section a more careful account of how the authors arrive at their conclusions is needed. For example, the authors state that "a large fraction of neurons were mixed-selective" but they only show or mention an analysis of two neurons (Figure 4E,F).

A few times in the text (line 19 in abstract, line 43, line 160, line 363), reference was made to the amounts of SFA observed in the brain or in particular regions, or the properties of neurons with SFA. These statements should have citations of actual studies (in the main text) to support them. If any of these claims are based on the author's own analysis of the Allen Institute data, they may want to include that in the Methods or Supplement and cite their Methods or Supplement section in the main text instead of citing the database.

It should be explicitly stated which synapses were being trained (recurrent weights, readout weights, or both) and there should be an explicit expression for output sigmoid neurons in the Methods.

An explanation or rationale for why SFA networks only had SFA in a portion of LIF neurons would be helpful.

One of the results I found most striking was the finding that only 1 network level with SFA was needed to achieve the 12AX task rather than the canonical 2 network levels. Given that the authors present the first instance of a SNN solving the 12AX task, it seems that these results are worth unpacking in greater depth in the Results and Discussion sections.

Figure 1: Perhaps include STORE and RECALL in schematic in C, or a general command module that represents analogous commands from the various tasks (i.e. duplicate, replicate, etc.).

Figure 2: The caption refers to "grey values" but it was not clear what/where exactly those are in the context of this figure. This should be clarified.

Figure 3: I found the Top to Bottom listing in the caption to be confusing because it did not refer to the labels on the figure very directly, leaving things open to interpretation. The caption should be re-written to clarify this.

Figure 4: Error in number of neurons (B-F)? Caption says 279 neurons but the main text says there were 320 neurons. Please clarify the discrepancy.

The plots in B and C should have annotations that orient the reader to the relevant parts of the task. For example, a line down the middle of Figure 4B indicating the start of duplication or reversal. In Figure 4C, it would be helpful to see where the abcde sequence falls on the time axes.

Line 52: Fairhall and others have written several reviews that could be cited here as well (Weber and Fairhall, 2019, for example).

Line 122: Replace "perfectly reproduced" with "accurately reproduced" since the output displayed in the figure is clearly not a "perfect" reproduction of the input even if it is deemed an accurate representation by whatever tolerance metric is used.

Line 129: It was confusing for me to determine which part of Figure 2 was the 3rd to last row. I suggest either labeling the rows in Figure 2 with A,B,C, etc or referring to the labels that are already there when describing a given row in the text, i.e. "The row labelled Thresholds in Figure 2 shows the temporal dynamics…".

Paragraph on line 146: I suggest re-wording this paragraph to make it clearer and more focused on your own results. After the first sentence, you should directly describe what you did to test whether the memory storage is supported or not by attractor dynamics. The discussion of the Wolff study and how it relates to your results should be incorporated into the Discussion section instead.

Line 166: I think you meant "upper five rows" not "upper four rows". In either case, I found it confusing to refer to the rows described in the text. This paragraph would be more readable if the results were relayed more directly. For example, the sentence that starts on line 166 could be re-written as: "Four fixed time constants were tested for the SNN with SFA (tau_a = 200 ms, 2 s, etc., see Table 1)."

Line 299: It is not clear how Figure 4B relates to this sentence. The citation to Figure 4B may be misplaced, or the sentence may need to be rewritten. The rest of this paragraph does not make a clear point and should be re-written.

Paragraph starting line 342 refers to points A, B, and C in Figure 2, but there are no such labels in that figure.

Line 349: Replace "currently not in the focus of attention" with "not attended".

Line 363: Given that this study did not examine the extent of SFA needed to improve performance, I suggest softening this statement. There could be a nonlinear relationship between the amount of SFA and performance on temporal computing tasks – perhaps too much SFA could diminish performance.

Line 365: This sentence should be rewritten. As it is written now, it indicates that the analysis itself was loose, but I think the intention was to point out that a strict alignment between the SFA and the task time scales is not necessary.

*Reviewer #2 (Recommendations for the authors):*

Salaj et al., simulate biophysically-inspired spiking neural networks that solve a range of sequential processing tasks. Understanding how neural networks perform sequential tasks is important for understanding how animals recognize patterns in time or perform sequential behaviors. Such processes are especially relevant for understanding language processing in humans. The authors show that spike-frequency adaptation can endow spiking networks with a form of working memory that allows them to solve sequential tasks.

In spike-frequency adaptation (SFA), neurons that have been active recently become harder to activate in the future, emitting fewer spikes for a given input.

Salaj et al., offers a proof-of-principle that spiking neural networks with SFA can support a dynamic memory, comparable to the rate-based Long Short-Term Memory (LSTM) networks from machine learning. Because of spike-frequency adaptation, neurons that have been active recently become harder to re-activate. A network can query a memory by checking which neurons remain silent upon recall. This model offers a compelling hypothesis for a form of short-term working memory that they term "negative imprinting".

As a computational study, this work has very few weaknesses. The model is simplified to the point where many components are biologically implausible, but this avoids excess detail that would make the results hard to interpret. Care is taken to incorporate spike-frequency adaptation in an abstract way that provides insight, but avoids strong assumptions. The memory capabilities and mechanisms that emerge in the trained models imply some testable predictions about features that one might find in biological networks.

However, the grounding of the model in biology is limited. This is understandable, since it is difficult to constrain a model using datasets that do not yet exist. The paper does not explore whether the statistics of spiking activity that emerges in the simulations is consistent with any of the operating regimes of spiking networks observed in experiments. The hypothesis is interesting, but it is likely that the models exhibit some unrealistic behaviors. It would be useful to discuss the implications of these differences. More specific, experimentally-testable predictions that could falsify the model would be welcome.

In the context of machine learning and neural computation, Salaj et al., offer empirical evidence that networks with spike-frequency adaptation can compete with the LSTM networks on simple sequential tasks. However, it remains unclear whether the strategies learned in spiking networks have a deep connection to LSTMs, or whether these results scale to harder problems. But, there is no need to solve everything at once. Salaj et al., address several technical problems in constructing such networks, which will be of interest to researchers exploring similar models.

Overall, this study asked whether spike-frequency adaptation could provide a computational substrate for sequential processing. Salaj et al., show, through a range of simulations on diverse tasks, that the answer is yes (at least in theory). This provides a clear modeling foundation that can guide further experimental and modeling studies.

However, this modeling framework might be in some sense too powerful. Spike-frequency adaptation endows neurons with a slow variable. Training via backpropagation-through-time allows networks to use these slow variables for sequential memory, but it is not clear that this is unique to spike-frequency adaptation. Any slow process could be similarly harnessed, provided it is affected by the history of spiking activity, and alters a neuron's computational properties.

An important question, then, is whether biological learning rules build networks that use spike-frequency adaptation in a way that is computationally similar to these simulations. This is not a question that can be answered through simulation alone, but the principles outlined in Salaj et al. suggest experiments that could address it.

Salaj et al., borrow concepts from machine learning to construct interpretable models of spiking neural computation. Their simulated networks solve sequential tasks using spiking neurons. It is not clear that their work applies directly to biological neural networks, but it provides concrete hypotheses. This work extends our theoretical understanding of potential roles for spike-frequency adaptation in neural computation. Previous work has indicated that spike-frequency adaptation can store an 'afterimage' of recent neural activity. Some models propose that this could allow neurons to optimize communication, by sending fewer spikes for persistent inputs. Previous studies have conjectured that similar mechanisms could support sequential processing. Salaj et al., stands as an important contribution toward making such hypotheses concrete, and positing specific mechanisms.

I was pleased to receive your manuscript for review. I found the results intriguing, and liked the paper. As a computational study, the work is very impressive and solid. However, I think the presentation must be revised to reach *eLife*'s target audience.

My understanding is that *eLife* targets a general audience in biology. Your work uses modeling to make a concrete hypothesis about the function of spike-frequency adaptation in neural computation. As written, it is relevant and accessible to theorists working at the intersection of neurophysiology, neural computation, and machine learning. Would it be worth elaborating on parts of the manuscript to reach out to an even broader audience of experimentalists in biology? Perhaps:

– Better orient readers as to what biophysical processes the paper addresses in the introduction.

– Discuss a specific compelling experiment that suggests a computational role for SFA, what specifically was measured, etc.

– Draw more specific connections to biology when possible.

– Highlight which aspects of the model have a clear biological interpretation.

– Be explicit about unphysiological aspects of the model (and why these assumptions are ok).

– Be critical of the simulations: are there any unphysiological behaviors? Should we be concerned?

– What ’specific’ experimental results could falsify your model? What would proving your model false imply?

– What core predictions of the model will likely survive, even if many of the details are wrong?

– Computational mechanisms are very flexible: there are likely other models that could show something similar. Is there something special about this solution that makes it more plausible than others?

Especially toward the end, further discussing the theory in the context of neurophysiology, and proposing a few concrete experiments (perhaps extending ones in prior literature), could be useful.

The library that supports these simulations is provided, but the code to generate the specific models in the paper is not. (Is this correct?) If so, is there any way to provide this? I realize research code is hard to clean up for publication, but for this study it would be welcome. It doesn't need to be a detailed tutorial, just some scripts or ipython notebooks that reproduce the major findings. (If you used notebooks, PDF snapshots of the executed notebooks might also do, if the code is visible?)

You show that a SVM cannot decode a memory trace during the delay/hold period from spiking activity. Can you also show that the SVM ‘can’ decode the memory, if it is also provided the slow adaptation variables? This will better confirm that it is spike-frequency adaptation that stores the memory trace.

Google Speech Commands are mentioned in the data availability statement and supplement, but nowhere in the main text. Can you discuss why these experiments were performed and what they tell us? Or, are they really necessary for the main result?

The "negative imprinting principle" is a major organizing concept introduced in this paper. At present, it is mentioned (with little explanation) in the introduction. The section introducing "negative imprinting" in the results is isolated. The idea is very general, and could apply to any process with slow timescales that depends on spiking history and affects neuronal dynamics. Should we simply think of "negative imprinting" as "firing rate suppression stores (some sort of?) memory trace". Is there any deeper connection with the components of a LSTM? Can you elaborate or speculate on further connections to other studies?

Can you address the concerns outlined in the public review, and above?

– Improve the presentation of the results to provide clearer and more concrete benefits to a broader audience in biology.

– Discuss limitations: nonphysical aspects of the model, and nonphysical behavior of the mode.

– Be as concrete as possible about what would be needed to falsify the prediction, ideally proposing experiments (at least abstractly).

– Discuss likely failure modes of the theory, and what this would imply for neurophysiology/neural computation.

– Discuss whether the backpropagation training is too powerful (Sejnowski lab applied it to neuronal parameters, and it learned slow synapses, so I think it really is: it uses anything it can to build a memory).

More generally, the text could benefit from a rewrite to improve organization and better guide readers. There are also some parts that are hard to read, but I'm less worried about those, as they can be caught in the final copyediting, etc.

Literature connections

Connections with the literature could be better explored. Discussing these might also make the paper more accessible to a general audience, and increase impact. Here are some I can think of:

GJ Gutierrez, S Denève. (2019). Population adaptation in efficient balanced networks. e*Life*

Kim, R., Li, Y., and Sejnowski, T. J. (2019). Simple framework for constructing functional spiking recurrent neural networks. Proceedings of the national academy of sciences, 116(45), 22811-22820.

Li Y, Kim R, Sejnowski TJ. (2020). Learning the synaptic and intrinsic membrane dynamics underlying working memory in spiking neural network models. bioRxiv.

The connection to Gutierrez and Denève, (2019) is interesting. In their model, they use spike-frequency adaptation to implement efficient coding: in principle, neurons could reduce their firing rates for persistent stimuli. This also creates a "negative imprint", and introduces a sort of sequential dependence in the neural code that is used to improve its efficiency (reduce the number of spikes needed).

Contrasting your work with Kim et al., (2019) might be useful. They configure the time-constants and population rates so that the network occupies a rate-coding regime. The explicit treatment of SFA in your work counters this: continuous, slow-timescale variables can be stored within neurons; there is no need to create a rate-coding regime to support a slow dynamical system.

Li et al., (2020) applied machine learning to adjust neuronal time constants to create slow timescales. Their model seems to have built working memory by slowing down synaptic integration. I'm not sure this is plausible, especially since the balanced state should be associated with high conductances which reduce the membrane time constant? Internal variables that support spike-frequency adaptation might be a more robust substrate for working memory.

Also, Jing Cai's 2021 Cosyne talk is relevant. (video is online). They trained deep semantic nets, which learn neuronal responses that look like language neurons in the brain. They argue that some neurons involved in language processing have activity that can be interpreted as a predictive model. This seems to point to a larger body of research on the neural correlates of language processing that might be worth discussing. I don't know if Cai's work is directly relevant, but any work on understanding how neurons process language would be very nice to discuss in context.

Perhaps point out: your work provides a neural substrate for working memory that is much more efficient than any of the attractor-based methods developed in neural mass/field theory in the past 30 years. Is all of that work on manifold attractors and neural assemblies nonsense? What about Mark Goldman's various tricks for getting slow dynamics in rate networks (non-normal dynamics, feedback control). SFA more efficient, is there any role for these other mechanisms? Population dynamics in frontal and parietal cortex do seem more consistent with mechanisms for working memory that use reverberating population activity. Can these other models do things that would be hard for implement using SFA? Or is SFA secretly an important factor in stabilizing these working-memory correlates that we observe in large populations?

Comments on the text

192: See section 2 of the Supplement for more results with sparse connectivity, enforcement of Dale's Law and comparison to ANNs.

This is very brief. When reading the main results, I was worried that these results might be incompatible with the operating regimes of biological networks. It is helpful to know whether the model still works with more realistic network. I think it is also important to examine the statistics of population activity and see if they deviate in any obvious way from neural recordings. Is this possible?

40: SFA denotes a feature of spiking neurons where their preceding firing activity transiently increases their firing threshold.

Surely SFA refers to a broad class of mechanisms that reduce firing rate after prolonged activity, not just threshold adaptation? Gain modulation, synaptic depression, other slow changes in channel states, could lower firing rates by adjusting other physiological parameters. But, in this work threshold adaptation is used as a qualitative proxy for a range of these phenomena. I'm curious, is there any broad survey of different types of spike-frequency adaptation? Would you expect different mechanisms to have different implications for computation?

66 including the well-known 12AX task

What is 12AX? (I had never heard of it.)

70: A practical advantage of this simple model is that it can be very efficiently simulated and is amenable to gradient descent training methods.

Is there a reference that could go here for further reading on simulation/training?

72: It assumes that the firing threshold A(t) of a leaky integrate-and-fire (LIF) neuron contains a variable component a(t) that increases by a fixed amount after each of its spikes z(t).

I suspect this introduction of the LIF model is too fast for a general audience. Some experimentalists (especially students) may not have heard of the LIF model before. Equation 1 is too abrupt. Define the model equations (in Methods) here. This is also important for making it clear what variable abbreviations mean.

88: We used Backpropagation through time (BPTT) for this, which is arguably the best performing optimization method for SNNs that is currently known.

This needs a citation to support it. Please provide a citation that (1) provides a good introduction to BPTT for a general audience, and (2) justifies that this is the best-performing method known.

Figure 1-C:

The implication is that this is a purely excitatory recurrent network; So there are no I cells and we should not think of this as similar to the balanced networks of Deneve, Machens, and colleagues, nor should we think of this as an inhibition-stabilized network in the stabilized supralinear regime? Should we believe that the operating regime of this network is at least vaguely similar to something that happens in some biological neural networks?

211: The 12AX task – which can be viewed as a simplified version of the Wisconsin Card Sorting task (Berg, 1948) – tests the capability of subjects to apply dynamically changing rules for detecting specific subsequences in a long sequence of symbols as target sequences, and to ignore currently irrelevant inputs (O'Reilly and Frank, 2006; MacDonald III, 2008).

I'm unfamiliar with the Wisconsin Card Sorting task is. The authors seem to assume that readers will already be familiar with common sequential tasks used in psychophysics. For a general audience, more background should be provided.

Figure 3:

I like this figure a lot; It makes it easy to grasp what the 12AX task is and how the spiking net solves it. The inconsistent font sizes and text orientations for the vertical axes is triggering some moderate dyslexia, however.

270: Obviously, this question also lies at the heart of open questions about the interplay between neural codes for syntax and semantics that enable language understanding in the human brain.

This is one of the more interesting connections; Perhaps it can be hinted at earlier as well. I suspect there is more recent work that one could cite exploring this. (and this was not obvious to me)

Page 9:

The results, as written, are a bit disorienting. Parts of the results seem to flow into discussion, then switch abruptly back to more results. It seems like sections could be better organized and more clearly linked into a single linear story. Some additional text could be provided to guide the reader throughout?

277: In particular, they also produce factorial codes, where separate neurons encode the position and identity of a symbol in a sequence.

I feel like I missed this result, could it be more explicitly emphasized wherever it is presented in the preceding text?

295: For comparison, we also trained a LIF network without SFA in exactly the same way with the same number of neurons. It achieved a performance of 0.0%.

This is really nice (and nicely written!). Could the simulations presented earlier benefit from a similar comparison?

However! At this point I'm very impressed, but now wondering: is this too powerful? Can backpropagation simply learn to harness any slow variables that might be present? Should we be worried about whether a backpropagation-trained network can be clearly compared to biology?

297: A diversity of neural codes in SNNs with SFA…

Is this section still talking about the repeat/reverse task?

322: Previous neural network solutions for similarly demanding temporal computing tasks were based on artificial Long Short-Term Memory (LSTM) units. These LSTM units are commonly used in machine learning, but they cannot readily be mapped to units of neural networks in the brain.

This is nice; More citations; Also, hint at this in the introduction? It's a good hook.

326: We have shown that by adding to the standard model for SNNs one important feature of a substantial fraction of neurons in the neocortex, SFA, SNNs become able to solve such demanding temporal computing tasks.

Can we conjecture (1) whether this in some way implements anything similar to the gates in LSTMs? and (2) whether we should expect to see anything like this in any biological network?

Methods, lines 429-235 are key:

This explains why we see scaling factors inside the Heaviside step function earlier; These do nothing for the actual network dynamics, but scale the pseudo-gradients in some useful way. The choice to attenuate the pseudo-gradients by $\γ=0.3$ is arbitrary and mysterious. It is possible to provide some intuition about why and how training fails when $\γ$ is too large? Is it possible to provide some intuition about why the pseudo-gradients should also be normalized by the current threshold of the neuron?

Presumably the role of the adaptation variable $a$ is also included in the BPTT derivatives? Or is it enough to adapt weights and leave the effect of this implicit? I ask because it seemed like Bellec, 2020 truncated the expansion of some of the partial derivatives, but training still worked.

What training approach was used for BPTT? In my own attempts, BPTT nearly always unstable because of successive loss of precision. Dissipative systems and a mixture of time constants on disparate scales are especially problematic. Any suggestions for us mortals?

---

## [Author Response]

Essential Revisions:1) The authors need to rewrite the results and introduction with a broad biological audience in mind. As it stands, it targets readers who work on artificial neural networks and the results are essentially benchmarks against artificial neural networks. Many of the assumptions and connections to biology are not discussed adequately. In particular, key omissions and simplifications need to be discussed: the artificial nature of training, the lack of structure in the recurrent network, the likely differences between the statistics of spiking in the model vs what might be observed in an intact brain.

We have rewritten the Introduction and Discussion, and also many parts of Results with a broad biological audience in mind. In particular, we have extended discussions about the relation of our work to neuroscience literature. We have changed some model parameters in order to reduce firing rates, and have added data on firing rates in our models. They appear to be now in a physiological range. In addition, we have carried out, for the 12AX task, control experiments that evaluate network performance in the presence of noise.

We have also added remarks regarding the lack of structure in the architecture of our models, in particular for the network that solves the 12AX task. We are pointing to the possibility that this may be a feature rather than a bug of our model: Cortical circuitry tends to be in our view much less structured than most handcrafted models. In particular, many simple assumptions about hierarchical processing in cortical networks tend to get into conflict with experimental data.

We have added further remarks that point out that this paper studies computational capabilities and limitations of different types of spiking neural networks, and not how these capabilities emerge in neural networks of the brain. We also have pointed to another recent paper (Bellec et al., Nature Communications) where it had been shown that for a variety of tasks and spiking neural network architectures the learning performance of BPTT can be approximated quite well by e-prop, which appears to be substantially more plausible learning method from the biological perspective. We have also added performance data for e-prop for one central task in this paper, the 12AX task. We also have pointed out at the end of the Discussion that temporal computing capabilities are likely to be to a large extent genetically encoded, rather than learned, and suggested for future work a concrete approach for testing whether this is in principle possible.

2) It would be helpful to lead the results with a simplified model that illustrates the negative imprinting mechanism directly. The results as they stand use rather complex models and provide largely observational evidence for the main claims of the paper. Since performance gains over other artificial neural networks are variable according to the task, some handle on when SFA is beneficial would help.

We agree, and hence we added a simplified model (new Figure 1D, E) that illustrates the negative imprinting principle directly, for a very simple temporal computing task. We have also added a number of control experiments where the same task was solved by different types of spiking neural networks, with and without SFA see in particular Figure 2B and C.

3) The relevance of SFA (as opposed to other slow timescale mechanisms) deserves fuller attention. The SFA characteristics are extracted from intracellular slice recordings where there is no substantial background activity. In an intact animal (with realistic spiking statistics) it is questionable whether the results will hold in a network where ongoing activity could maintain the SFA recovery variable close to steady-state. If this is the case, the conclusions need to be adequately tempered.

We agree, and have added a new section “Comparing the contribution of SFA to temporal computing with that of other slow processes in neurons and synapses” to Results.

Regarding the remark that the SFA variable could saturate in-vivo, due to bombardment by synaptic input: We have added a discussion of the implicit self-protection mechanism of the adapted state of neurons with SFA: Whenever information has been loaded into their hidden variable, the adaptive threshold, this automatically protects the neurons from firing in response to weaker inputs. But if these neurons do not fire, the value of their salient hidden variable is not affected by noise inputs. We have discussed this mechanism in particular in the context of another candidate mechanism for storing information in the hidden state of a neuron: spike triggered increases of neuronal excitability. This mechanism has no such self-protection against noise, and also the contribution of such neurons (“ELIF neurons”) to temporal computing is much less pronounced.

Reviewer #1 (Recommendations for the authors):The authors demonstrate the utility of spike frequency adaptation (SFA) in an artificial spiking neural network (SNN) for accomplishing tasks with a temporal dimension. They first show that a SNN with SFA can perform a simple store/recall task. They then test their model against tasks with increasing complexity. The authors find that in all of these tests, the SNN with SFA is able to match or outperform the most commonly implemented artificial neural network mechanisms which are also less biologically relevant. This study brings an awareness of the SFA mechanism to neural network modelers and it quantifies the improvement it brings to the performance of complex tasks. Furthermore, this study offers neurobiologists a new perspective on the SFA mechanism and its function in neural processing.The most exciting strength of this paper is its multidisciplinary nature. It has the potential to drive innovation in the field of artificial neural networks (ANNs) by exploring how a known biological neural mechanism – SFA – can impact the performance of ANNs for solving temporally-complex tasks.The multidisciplinary nature of this paper is a double-edged sword, however. At times there was a bit of a disconnect between the conclusions and the results. In most cases, the performance of the model against a canonical benchmark is rigorously quantified in a way that is standard in many neuroengineering papers. However, there were also multiple attempts to connect the model's performance to actual findings in the brain. I think that this is a worthwhile thing to do because it targets a broader readership that makes the study more relevant to the eLife audience; however, those comparisons were less rigorous than the ANN benchmark comparisons. This was particularly the case for the Negative Imprinting phenomenon and the emergent mixed-selectivity codes.With regards to the rigorous benchmark comparisons presented, this paper could be made more accessible to the general neuroscience audience by putting the results and benchmark comparisons into a fuller context for those who are not as familiar with how well recent models have performed on these tasks. At times, the results showed very impressive improvement of the model over standard benchmarks while for other tasks the improvements appeared incremental. It would be helpful to have the performance improvements from the SFA model placed in the context of recent improvements in the canonical or state-of-the-art ANNs. This would allow the reader to know whether a small increase in performance represents an incremental improvement or a giant leap forward in the context of the history of ANNs.Overall, there are a few main things I suggest the authors focus on revising. The figures and their captions could be made clearer in some places, as well as places in the main text that refer to specific parts of the figures. The other areas I think the authors should focus on are the last section of the Results, the Negative Imprinting Principle section, and the Discussion section.The last section of the Results should probably be rewritten – especially the first paragraph. In general, in this section a more careful account of how the authors arrive at their conclusions is needed. For example, the authors state that "a large fraction of neurons were mixed-selective" but they only show or mention an analysis of two neurons (Figure 4E,F).

The first paragraph is rewritten (a few sentences reordered). More details (what one can see from Figure 4) are given (L416-420).

Analysis that we conducted to conclude about mixed-selectivity of neurons was described, but a reference to Figure 4D that shows fractions of neurons selective to different combinations of variables was missing, and now added (L428-434). More details about the analysis are provided in Methods, last paragraph.

A few times in the text (line 19 in abstract, line 43, line 160, line 363), reference was made to the amounts of SFA observed in the brain or in particular regions, or the properties of neurons with SFA. These statements should have citations of actual studies (in the main text) to support them. If any of these claims are based on the author's own analysis of the Allen Institute data, they may want to include that in the Methods or Supplement and cite their Methods or Supplement section in the main text instead of citing the database.

We added a new Appendix figure 8 to the Appendix 1 and referenced it in the introduction (L37) to make this finding more clear. We will also be publishing the code for the reproduction of this figure based on the Allen Institute data.

It should be explicitly stated which synapses were being trained (recurrent weights, readout weights, or both) and there should be an explicit expression for output sigmoid neurons in the Methods.

The Methods are updated to explicitly state the trained synapses (L669). The expressions for the sigmoid (used in 1D and 20D STORE-RECALL task) and softmax function (used in sMNIST, 12AX, Duplication/Reversal tasks) are added (L645).

An explanation or rationale for why SFA networks only had SFA in a portion of LIF neurons would be helpful.

We added remarks regarding this decision (to test the effect on the performance) and new results demonstrating the effect of the proportion of neurons with SFA in the network on the performance (L355-362).

One of the results I found most striking was the finding that only 1 network level with SFA was needed to achieve the 12AX task rather than the canonical 2 network levels. Given that the authors present the first instance of a SNN solving the 12AX task, it seems that these results are worth unpacking in greater depth in the Results and Discussion sections.

We included a paragraph discussing this (L485-494).

Figure 1: Perhaps include STORE and RECALL in schematic in C, or a general command module that represents analogous commands from the various tasks (i.e. duplicate, replicate, etc.).

We added a simplified model for illustration of a simple version of the STORE-RECALL task, (new schematic in Figure 1D), and we also use it to explain the negative imprinting principle.

Figure 2: The caption refers to "grey values" but it was not clear what/where exactly those are in the context of this figure. This should be clarified.

The caption of Figure 2 has been adapted to avoid confusion.

Figure 3: I found the Top to Bottom listing in the caption to be confusing because it did not refer to the labels on the figure very directly, leaving things open to interpretation. The caption should be re-written to clarify this.

Some labels in Figure 3 were vertical, some horizontal. This is changed now (everything horizontal), and the caption is slightly adapted.

Figure 4: Error in number of neurons (B-F)? Caption says 279 neurons but the main text says there were 320 neurons. Please clarify the discrepancy.The plots in B and C should have annotations that orient the reader to the relevant parts of the task. For example, a line down the middle of Figure 4B indicating the start of duplication or reversal. In Figure 4C, it would be helpful to see where the abcde sequence falls on the time axes.

A part stating that some neurons were discarded from the analysis is added in the caption. The criteria to remove neurons was described in the last section of Materials and methods (L860).

Figure 4B and C are modified as suggested.

Line 52: Fairhall and others have written several reviews that could be cited here as well (Weber and Fairhall, 2019, for example).

Citations added in the paragraph discussing efficient neural coding (L43-51).

Line 122: Replace "perfectly reproduced" with "accurately reproduced" since the output displayed in the figure is clearly not a "perfect" reproduction of the input even if it is deemed an accurate representation by whatever tolerance metric is used.

Suggestion implemented (L214).

Line 129: It was confusing for me to determine which part of Figure 2 was the 3rd to last row. I suggest either labeling the rows in Figure 2 with A,B,C, etc or referring to the labels that are already there when describing a given row in the text, i.e. "The row labelled Thresholds in Figure 2 shows the temporal dynamics…".

In the previous version, we had referred to the thresholds in Figure 2 to explain the negative imprinting principle. This is improved now – we describe and explain the negative imprinting principle by referring to Figure 1E (L141-163).

Paragraph on line 146: I suggest re-wording this paragraph to make it clearer and more focused on your own results. After the first sentence, you should directly describe what you did to test whether the memory storage is supported or not by attractor dynamics. The discussion of the Wolff study and how it relates to your results should be incorporated into the Discussion section instead.

Since the reference to Wolff study is the motivation for that experiment, we did not find it appropriate to remove it from this section (Results). However, the paragraph is rewritten (L190-206), stating that the main result is consistent with the experimental data, and all the details are described in Materials and methods (L749).

The Wolff study is discussed again in the Discussion (L456).

Line 166: I think you meant "upper five rows" not "upper four rows". In either case, I found it confusing to refer to the rows described in the text. This paragraph would be more readable if the results were relayed more directly. For example, the sentence that starts on line 166 could be re-written as: "Four fixed time constants were tested for the SNN with SFA (tau_a = 200 ms, 2 s, etc., see Table 1)."

Suggestion implemented (L227).

Line 299: It is not clear how Figure 4B relates to this sentence. The citation to Figure 4B may be misplaced, or the sentence may need to be rewritten. The rest of this paragraph does not make a clear point and should be re-written.

The relation of the sentence to Figure 4B is rewritten, by explicitly pointing to the parts of the figure (L416-425). The paragraph and its relation to Figure 4B-F is improved (L417).

Paragraph starting line 342 refers to points A, B, and C in Figure 2, but there are no such labels in that figure.

This was a misunderstanding of the reader (we referred to time points A, B, C, and also to Figure 2), but we removed the references to these time points A, B, C, and rephrased the section (L464).

Line 349: Replace "currently not in the focus of attention" with "not attended".

Modification implemented (L457).

Line 363: Given that this study did not examine the extent of SFA needed to improve performance, I suggest softening this statement. There could be a nonlinear relationship between the amount of SFA and performance on temporal computing tasks – perhaps too much SFA could diminish performance.

Suggestion implemented, (paragraph starting on L562).

We also included a new result in the section for the 12AX where we demonstrate the effect that the amount of SFA (fraction of neurons with SFA) has on the performance (L355-362), and state the result where too much SFA diminished the performance (L356).

Line 365: This sentence should be rewritten. As it is written now, it indicates that the analysis itself was loose, but I think the intention was to point out that a strict alignment between the SFA and the task time scales is not necessary.

Suggestion implemented (L569-571).

Reviewer #2 (Recommendations for the authors):Salaj et al., simulate biophysically-inspired spiking neural networks that solve a range of sequential processing tasks. Understanding how neural networks perform sequential tasks is important for understanding how animals recognize patterns in time or perform sequential behaviors. Such processes are especially relevant for understanding language processing in humans. The authors show that spike-frequency adaptation can endow spiking networks with a form of working memory that allows them to solve sequential tasks.In spike-frequency adaptation (SFA), neurons that have been active recently become harder to activate in the future, emitting fewer spikes for a given input.Salaj et al., offers a proof-of-principle that spiking neural networks with SFA can support a dynamic memory, comparable to the rate-based Long Short-Term Memory (LSTM) networks from machine learning. Because of spike-frequency adaptation, neurons that have been active recently become harder to re-activate. A network can query a memory by checking which neurons remain silent upon recall. This model offers a compelling hypothesis for a form of short-term working memory that they term "negative imprinting".As a computational study, this work has very few weaknesses. The model is simplified to the point where many components are biologically implausible, but this avoids excess detail that would make the results hard to interpret. Care is taken to incorporate spike-frequency adaptation in an abstract way that provides insight, but avoids strong assumptions. The memory capabilities and mechanisms that emerge in the trained models imply some testable predictions about features that one might find in biological networks.However, the grounding of the model in biology is limited. This is understandable, since it is difficult to constrain a model using datasets that do not yet exist. The paper does not explore whether the statistics of spiking activity that emerges in the simulations is consistent with any of the operating regimes of spiking networks observed in experiments. The hypothesis is interesting, but it is likely that the models exhibit some unrealistic behaviors. It would be useful to discuss the implications of these differences. More specific, experimentally-testable predictions that could falsify the model would be welcome.In the context of machine learning and neural computation, Salaj et al., offer empirical evidence that networks with spike-frequency adaptation can compete with the LSTM networks on simple sequential tasks. However, it remains unclear whether the strategies learned in spiking networks have a deep connection to LSTMs, or whether these results scale to harder problems. But, there is no need to solve everything at once. Salaj et al., address several technical problems in constructing such networks, which will be of interest to researchers exploring similar models.Overall, this study asked whether spike-frequency adaptation could provide a computational substrate for sequential processing. Salaj et al., show, through a range of simulations on diverse tasks, that the answer is yes (at least in theory). This provides a clear modeling foundation that can guide further experimental and modeling studies.However, this modeling framework might be in some sense too powerful. Spike-frequency adaptation endows neurons with a slow variable. Training via backpropagation-through-time allows networks to use these slow variables for sequential memory, but it is not clear that this is unique to spike-frequency adaptation. Any slow process could be similarly harnessed, provided it is affected by the history of spiking activity, and alters a neuron's computational properties.An important question, then, is whether biological learning rules build networks that use spike-frequency adaptation in a way that is computationally similar to these simulations. This is not a question that can be answered through simulation alone, but the principles outlined in Salaj et al. suggest experiments that could address it.Salaj et al., borrow concepts from machine learning to construct interpretable models of spiking neural computation. Their simulated networks solve sequential tasks using spiking neurons. It is not clear that their work applies directly to biological neural networks, but it provides concrete hypotheses. This work extends our theoretical understanding of potential roles for spike-frequency adaptation in neural computation. Previous work has indicated that spike-frequency adaptation can store an 'afterimage' of recent neural activity. Some models propose that this could allow neurons to optimize communication, by sending fewer spikes for persistent inputs. Previous studies have conjectured that similar mechanisms could support sequential processing. Salaj et al., stands as an important contribution toward making such hypotheses concrete, and positing specific mechanisms.I was pleased to receive your manuscript for review. I found the results intriguing, and liked the paper. As a computational study, the work is very impressive and solid. However, I think the presentation must be revised to reach eLife's target audience.My understanding is that eLife targets a general audience in biology. Your work uses modeling to make a concrete hypothesis about the function of spike-frequency adaptation in neural computation. As written, it is relevant and accessible to theorists working at the intersection of neurophysiology, neural computation, and machine learning. Would it be worth elaborating on parts of the manuscript to reach out to an even broader audience of experimentalists in biology? Perhaps:– Better orient readers as to what biophysical processes the paper addresses in the introduction.– Discuss a specific compelling experiment that suggests a computational role for SFA, what specifically was measured, etc.– Draw more specific connections to biology when possible.– Highlight which aspects of the model have a clear biological interpretation.– Be explicit about unphysiological aspects of the model (and why these assumptions are ok).– Be critical of the simulations: are there any unphysiological behaviors? Should we be concerned?– What ’specific’ experimental results could falsify your model? What would proving your model false imply?– What core predictions of the model will likely survive, even if many of the details are wrong?– Computational mechanisms are very flexible: there are likely other models that could show something similar. Is there something special about this solution that makes it more plausible than others?

We have added a new section with results comparing SFA to other mechanisms (L270-299).

We also discussed the self-protection mechanism of memory content (L532-539) achieved through the means of SFA (but also, other depressing mechanisms).

Especially toward the end, further discussing the theory in the context of neurophysiology, and proposing a few concrete experiments (perhaps extending ones in prior literature), could be useful.

Suggestion implemented. The Discussion section is rewritten, and we make a few suggestions for the experiments relating to the existing ones in prior literature (L458-463, L540-546, L557-561, L562-565).

The library that supports these simulations is provided, but the code to generate the specific models in the paper is not. (Is this correct?) If so, is there any way to provide this? I realize research code is hard to clean up for publication, but for this study it would be welcome. It doesn't need to be a detailed tutorial, just some scripts or ipython notebooks that reproduce the major findings. (If you used notebooks, PDF snapshots of the executed notebooks might also do, if the code is visible?)

We are in the process of preparation and plan to release all code to reproduce all results presented in the paper and Appendix 1.

You show that a SVM cannot decode a memory trace during the delay/hold period from spiking activity. Can you also show that the SVM ‘can’ decode the memory, if it is also provided the slow adaptation variables? This will better confirm that it is spike-frequency adaptation that stores the memory trace.

This is possible but is somewhat trivial. We attempted to clarify this better by adding a new result and Figure 1E where we illustrate the negative imprinting principle and how slow adaptation variables are exploited for the memory.

Google Speech Commands are mentioned in the data availability statement and supplement, but nowhere in the main text. Can you discuss why these experiments were performed and what they tell us? Or, are they really necessary for the main result?

We included two paragraphs: in the Results on (L252), and explained the reason to consider this task (L513-516).

The "negative imprinting principle" is a major organizing concept introduced in this paper. At present, it is mentioned (with little explanation) in the introduction. The section introducing "negative imprinting" in the results is isolated. The idea is very general, and could apply to any process with slow timescales that depends on spiking history and affects neuronal dynamics. Should we simply think of "negative imprinting" as "firing rate suppression stores (some sort of?) memory trace". Is there any deeper connection with the components of a LSTM? Can you elaborate or speculate on further connections to other studies?

We significantly expanded the negative imprinting section (L141) and added new results (also Fig1D, E) to illustrate the negative imprinting principle better. We also expanded the Discussion section with a paragraph on the implications of our results (L520-546), and proposed a testable hypothesis (L458-463, L476-480). The relation to LSTM is discussed in more detail at another point in the rebuttal below, but mentioned in the Discussion (L516-519).

Can you address the concerns outlined in the public review, and above?– Improve the presentation of the results to provide clearer and more concrete benefits to a broader audience in biology.

Introduction section is expanded (e.g., L43-51), Discussion section is rewritten, and the connection to neuroscience literature is discussed (L478, L528-546, L562-565).

– Discuss limitations: nonphysical aspects of the model, and nonphysical behavior of the mode.

The LIF model, in principle, can produce neuronal activity in an unphysiological range. We state the average firing rates for our experiments, and they appear to be in a meaningful range.

In the text, we state that the firing rate of neurons in a physiological range of sparse activity is needed if we want to compare them to brain activity, and implications for understanding the brain (L506-512).

– Be as concrete as possible about what would be needed to falsify the prediction, ideally proposing experiments (at least abstractly).– Discuss likely failure modes of the theory, and what this would imply for neurophysiology/neural computation.

Our model makes a number of concrete suggestions for further experiments which can validate the role of SFA for temporal computation. It suggests that a refined decoder that takes negative imprinting into account is able to elucidate the transformation of stored information between time points A (encoding) and an intermediate time point C (network reactivation) in the experiment of (Wolff et al., 2017). A more direct approach would be to reproduce the results of (Wolff et al., 2017) but with calcium imaging. If one can separately tag neurons with SFA with genetic mechanisms, one could see their particular role within temporal computing tasks. Another approach would be an ablation study where SFA is disabled via optogenetic manipulation, for which our results predict that the performance of such networks on working memory tasks would degrade.

In discussion, we state that the lack of the mathematical models that describe the processes on slower time scales are missing (L540-546). This could, in principle, be used as an experiment to support or contradict our results.

We added the remarks on the implications of our results (L520-539).

– Discuss whether the backpropagation training is too powerful (Sejnowski lab applied it to neuronal parameters, and it learned slow synapses, so I think it really is: it uses anything it can to build a memory).

We added a new section with results (L270-299) where we compare SFA to other mechanisms (see Figure 2 B,C) which also demonstrates that backpropagation is not too powerful and without a capable substrate (like SFA) cannot solve the tasks. Also, we did not train any of the time constants or similar parameters which could change the temporal dynamics of the models.

More generally, the text could benefit from a rewrite to improve organization and better guide readers. There are also some parts that are hard to read, but I'm less worried about those, as they can be caught in the final copyediting, etc.Literature connectionsConnections with the literature could be better explored. Discussing these might also make the paper more accessible to a general audience, and increase impact. Here are some I can think of:GJ Gutierrez, S Denève. (2019). Population adaptation in efficient balanced networks. ELife

We have added the suggested reference (L43-51, L71).

Kim, R., Li, Y., and Sejnowski, T. J. (2019). Simple framework for constructing functional spiking recurrent neural networks. Proceedings of the national academy of sciences, 116(45), 22811-22820.

This is one of many papers presenting a novel method of converting rate networks to spiking networks working in a rate regime. Since our work is not about learning methods in spiking networks, we do not find it appropriate to discuss this specific algorithm.

Li Y, Kim R, Sejnowski TJ. (2020). Learning the synaptic and intrinsic membrane dynamics underlying working memory in spiking neural network models. bioRxiv.

This paper also discusses the learning method and learning the time constants, none of which are related to the results or message of our paper. Our conclusion is that this paper is also not appropriate related literature for our work. However, we have added more recent neuroscience-related literature regarding working memory (L537): (Mongillo et al., 2018; Kim and Sejnowski, 2021).

The connection to Gutierrez and Denève, (2019) is interesting. In their model, they use spike-frequency adaptation to implement efficient coding: in principle, neurons could reduce their firing rates for persistent stimuli. This also creates a "negative imprint", and introduces a sort of sequential dependence in the neural code that is used to improve its efficiency (reduce the number of spikes needed).

Discussed in Introduction (L43-51), but in the context of efficient, and stable neural codes. Referenced again after introducing the negative imprinting (L71).

Contrasting your work with Kim et al., (2019) might be useful. They configure the time-constants and population rates so that the network occupies a rate-coding regime. The explicit treatment of SFA in your work counters this: continuous, slow-timescale variables can be stored within neurons; there is no need to create a rate-coding regime to support a slow dynamical system.Li et al., (2020) applied machine learning to adjust neuronal time constants to create slow timescales. Their model seems to have built working memory by slowing down synaptic integration. I'm not sure this is plausible, especially since the balanced state should be associated with high conductances which reduce the membrane time constant? Internal variables that support spike-frequency adaptation might be a more robust substrate for working memory.Also, Jing Cai's 2021 Cosyne talk is relevant. (video is online). They trained deep semantic nets, which learn neuronal responses that look like language neurons in the brain. They argue that some neurons involved in language processing have activity that can be interpreted as a predictive model. This seems to point to a larger body of research on the neural correlates of language processing that might be worth discussing. I don't know if Cai's work is directly relevant, but any work on understanding how neurons process language would be very nice to discuss in context.

Interesting work (and related, to some extent because of the similar approach to analyze the selectivity of neurons), but no publication or a preprint available yet.

Perhaps point out: your work provides a neural substrate for working memory that is much more efficient than any of the attractor-based methods developed in neural mass/field theory in the past 30 years. Is all of that work on manifold attractors and neural assemblies nonsense? What about Mark Goldman's various tricks for getting slow dynamics in rate networks (non-normal dynamics, feedback control). SFA more efficient, is there any role for these other mechanisms? Population dynamics in frontal and parietal cortex do seem more consistent with mechanisms for working memory that use reverberating population activity. Can these other models do things that would be hard for implement using SFA? Or is SFA secretly an important factor in stabilizing these working-memory correlates that we observe in large populations?Comments on the text192: See section 2 of the Supplement for more results with sparse connectivity, enforcement of Dale's Law and comparison to ANNs.This is very brief. When reading the main results, I was worried that these results might be incompatible with the operating regimes of biological networks. It is helpful to know whether the model still works with more realistic network. I think it is also important to examine the statistics of population activity and see if they deviate in any obvious way from neural recordings. Is this possible?

We added the firing rates of neurons in experiments (L183, L348, L410), and a new result testing the robustness of the trained model to noise in the input (L363-370) and more details in (Appendix, section 5).

40: SFA denotes a feature of spiking neurons where their preceding firing activity transiently increases their firing threshold.Surely SFA refers to a broad class of mechanisms that reduce firing rate after prolonged activity, not just threshold adaptation? Gain modulation, synaptic depression, other slow changes in channel states, could lower firing rates by adjusting other physiological parameters. But, in this work threshold adaptation is used as a qualitative proxy for a range of these phenomena. I'm curious, is there any broad survey of different types of spike-frequency adaptation? Would you expect different mechanisms to have different implications for computation?

We added a new section with results comparing the SFA to other mechanisms (L270).

However, the primary focus of our work was whether the SFA could implement working memory (through negative imprinting) and enable powerful temporal computations in SNNs (L61-75, L190-201, L456).

66 including the well-known 12AX taskWhat is 12AX? (I had never heard of it.)

We removed “well-known”, and expanded the section about this task with a couple of introductory sentences (L302-L306).

70: A practical advantage of this simple model is that it can be very efficiently simulated and is amenable to gradient descent training methods.Is there a reference that could go here for further reading on simulation/training?

We added a citation to (Bellec et al., 2018) (L99).

72: It assumes that the firing threshold A(t) of a leaky integrate-and-fire (LIF) neuron contains a variable component a(t) that increases by a fixed amount after each of its spikes z(t).I suspect this introduction of the LIF model is too fast for a general audience. Some experimentalists (especially students) may not have heard of the LIF model before. Equation 1 is too abrupt. Define the model equations (in Methods) here. This is also important for making it clear what variable abbreviations mean.

We have adapted the text to introduce the LIF model more gently (L99-103). However, our best judgment was not to move the equations for LIF to the main results.

88: We used Backpropagation through time (BPTT) for this, which is arguably the best performing optimization method for SNNs that is currently known.This needs a citation to support it. Please provide a citation that (1) provides a good introduction to BPTT for a general audience, and (2) justifies that this is the best-performing method known.

References for introduction of BPTT are added (L126, L657), and for the second part, we softened the statement (L88, L126).

Figure 1-C:The implication is that this is a purely excitatory recurrent network; So there are no I cells and we should not think of this as similar to the balanced networks of Deneve, Machens, and colleagues, nor should we think of this as an inhibition-stabilized network in the stabilized supralinear regime? Should we believe that the operating regime of this network is at least vaguely similar to something that happens in some biological neural networks?

Appendix 1 contains results of sparse networks respecting Dale’s law that are trained and in fact achieve even better results relative to the models not respecting these constraints (Appendix 1, section 2). Figure 1C does not imply that the models are purely excitatory. In fact, the weights of each synapse are freely learned under no constraint regarding its sign (L642).

211: The 12AX task – which can be viewed as a simplified version of the Wisconsin Card Sorting task (Berg, 1948) – tests the capability of subjects to apply dynamically changing rules for detecting specific subsequences in a long sequence of symbols as target sequences, and to ignore currently irrelevant inputs (O'Reilly and Frank, 2006; MacDonald III, 2008).I'm unfamiliar with the Wisconsin Card Sorting task is. The authors seem to assume that readers will already be familiar with common sequential tasks used in psychophysics. For a general audience, more background should be provided.

The section is expanded, and the comparison to the Wisconsin Card Sorting is removed (L302-306).

Figure 3:I like this figure a lot; It makes it easy to grasp what the 12AX task is and how the spiking net solves it. The inconsistent font sizes and text orientations for the vertical axes is triggering some moderate dyslexia, however.

Modified to be consistent.

270: Obviously, this question also lies at the heart of open questions about the interplay between neural codes for syntax and semantics that enable language understanding in the human brain.This is one of the more interesting connections; Perhaps it can be hinted at earlier as well. I suspect there is more recent work that one could cite exploring this. (and this was not obvious to me)

We updated the introduction to hint at this earlier (L83-87), however, we could not find more recent work to cite.

Page 9:The results, as written, are a bit disorienting. Parts of the results seem to flow into discussion, then switch abruptly back to more results. It seems like sections could be better organized and more clearly linked into a single linear story. Some additional text could be provided to guide the reader throughout?

This would be difficult to disentangle, because it was a motivation for such an analysis to conduct. Parts of the paragraph are rewritten for easier readability (page 12). We hint at this task and the neural codes that emerge and give references, (in the Introduction, (L83-87)), hence this paragraph should be easier to understand now.

277: In particular, they also produce factorial codes, where separate neurons encode the position and identity of a symbol in a sequence.I feel like I missed this result, could it be more explicitly emphasized wherever it is presented in the preceding text?

Introduced in (L385). By factorial code is meant that, for example, position and identity of an item are encoded separately by some neurons.

295: For comparison, we also trained a LIF network without SFA in exactly the same way with the same number of neurons. It achieved a performance of 0.0%.This is really nice (and nicely written!). Could the simulations presented earlier benefit from a similar comparison?

We also do that for the 12AX task (L358), STORE-RECALL and sMNIST task (Figure 2B, C, bar with the label LIF), also in Table 1 (first row).

However! At this point I'm very impressed, but now wondering: is this too powerful? Can backpropagation simply learn to harness any slow variables that might be present? Should we be worried about whether a backpropagation-trained network can be clearly compared to biology?

To address this issue we added a new Results section where we compare SFA to different mechanisms (L270), all trained with backpropagation. It is true that backpropagation can in theory harness any slow variables, however, it is important to point out that the time constants are not trained, and that some mechanisms, despite long time constants, are not able to be exploited to solve the working memory tasks (see these new results, L270).

297: A diversity of neural codes in SNNs with SFA…Is this section still talking about the repeat/reverse task?

We slightly modified the title for this part, by adding “trained to carry out operations on sequences”, (L413). It should be clearer now that it relates to the duplicate/reverse task.

322: Previous neural network solutions for similarly demanding temporal computing tasks were based on artificial Long Short-Term Memory (LSTM) units. These LSTM units are commonly used in machine learning, but they cannot readily be mapped to units of neural networks in the brain.This is nice; More citations; Also, hint at this in the introduction? It's a good hook.

We mention that LSTMs are state-of-the-art in machine learning (ML) (L516) and emphasize that machine learning commonly uses artificial models that are not biologically realistic (while we are able to do similar tasks with biologically realistic models).

326: We have shown that by adding to the standard model for SNNs one important feature of a substantial fraction of neurons in the neocortex, SFA, SNNs become able to solve such demanding temporal computing tasks.Can we conjecture (1) whether this in some way implements anything similar to the gates in LSTMs? and (2) whether we should expect to see anything like this in any biological network?

The similarity to the LSTMs is that both LSTM and SFA have a “slow variable”. However, that is where the similarity ends. The slow variable (cell state) of LSTM is completely controlled by the trained gates which control the manipulation of the state based on the input. It can be viewed as a random access memory. In contrast, the slow variable of SFA decays constantly to its baseline and is modified by the neuron output in a single direction, which makes it much less flexible compared to LSTM.

Thus, we can say (1) no, this does not implement anything similar to the LSTM gates. The only similarity is in the improved performance on the temporal computing tasks.

Regarding (2), yes, SFA is a prominent feature of biological networks and in the discussion (L458-463, L476-480), we discuss two hypotheses that could be used to verify if the SFA is used in the biological networks in the way we predict.

Methods, lines 429-235 are key:This explains why we see scaling factors inside the Heaviside step function earlier; These do nothing for the actual network dynamics, but scale the pseudo-gradients in some useful way. The choice to attenuate the pseudo-gradients by $\γ=0.3$ is arbitrary and mysterious. It is possible to provide some intuition about why and how training fails when $\γ$ is too large? Is it possible to provide some intuition about why the pseudo-gradients should also be normalized by the current threshold of the neuron?

We added a citation to Bellec et al., 2018 pointing to the fact that we used the same training method of that work and did not consider the effect of this parameter in this paper.

Presumably the role of the adaptation variable $a$ is also included in the BPTT derivatives? Or is it enough to adapt weights and leave the effect of this implicit? I ask because it seemed like Bellec, 2020 truncated the expansion of some of the partial derivatives, but training still worked.

We added new results that train the networks with the method of Bellec et al., 2020 (L92, L352-354, L550-557).

What training approach was used for BPTT? In my own attempts, BPTT nearly always unstable because of successive loss of precision. Dissipative systems and a mixture of time constants on disparate scales are especially problematic. Any suggestions for us mortals?

The code which implements this is already published by Bellec et al., 2018: https://github.com/IGITUGraz/LSNN-official

The code for this paper is in preparation and will be published alongside the paper.